# Intelligence and Moral Development: A Critical Historical Review and Future Directions

**DOI:** 10.3390/jintelligence13070072

**Published:** 2025-06-22

**Authors:** Frank Fair, Daniel Fasko

**Affiliations:** 1Department of Psychology and Philosophy, Sam Houston State University, Huntsville, TX 77341, USA; 2School of Educational Foundations Leadership and Policy, Bowling Green State University, Bowling Green, OH 43403, USA; dfasko@bgsu.edu

**Keywords:** intelligence, moral development, psychology, phronesis, neuroscience

## Abstract

This paper is a critical, historical review of the literature on intelligence and moral development. In this review we come to a number of conclusions. For example, we identify methodological issues in past research on intelligence in relation to moral development, from Wiggam’s paper in 1941 through the first quarter of the 21st century, and we commend research done with methodological improvements we specify. Also, we conclude that Heyes’ evolutionary psychology that humans have a specifiable “starter kit” of processes that produce “cognitive gadgets,” including those used in normative thinking, should be given further attention. But, importantly, we note that these “gadgets” may be “malware” or be missing. Another conclusion is that Gert’s account of harms and benefits, of the moral rules, of how the rules are justified, and of how violations are justified, can be a fruitful component of the study of moral development. Furthermore, we argue that the work on wisdom by Sternberg, Kristjansson, and others is important to grasp for its relevance to putting morality into action. Lastly, we discuss areas for future research, especially in neuroscience, and we recommend paying attention to practices for the building of practical wisdom and morality.

## 1. Introduction

This paper is a critical, historical review of literature on the relationship between intelligence and moral development. The main purpose of this review is to try to map out the relationship between two very large and diversely understood aspects of our human nature.

At the outset, in delving into this literature we noticed a number of methodological issues in many studies that appeared problematic, such as how the participants were selected and how intelligence was determined see the Conclusion in ([27]). We do offer suggestions about how to address those issues, but there were larger issues that needed attention. Hence, the following review has five main parts: Section 2, Intelligence from a “Starter Kit” to Mindware, Section 3, Intelligence and Moral Development: Multiple Perspectives and Classic Research, Section 4, a Different Take on Morality, and Section 5, Intelligence and Morality in Action: Wisdom and Wisdom-Building. Section 6 of the paper asks So Where Do We Go from Here? and contains a number of specific suggestions for future research.

## 2. Intelligence from a “Starter Kit” to Mindware

### 2.1. A Controversy

As many researchers, educators, and others involved in the field of intelligence know, there has been much controversy regarding the definition of intelligence over several centuries. In the past century, according to [13] ([13]), there is apparently no consensus definition of the term in literature. Interestingly, [36] ([36]) points out that the definition of “human” intelligence has benefitted from the methodology of psychometrically measuring intelligence. Fifteen years ago, [16] ([16]) noted that research on the brain has contributed to our understanding of intelligence. This contribution will be discussed later in this paper. The following is a brief survey of the variety of ways of understanding what intelligence is, and it also indicates which of those ways have seemed useful to us in trying to trace the relations between intelligence and moral development.

The main argument from the psychological perspective revolves around whether there is a single general intelligence or whether there are perhaps multiple forms of intelligence, a claim developed, for example, by [12]’s ([12]) fluid and crystallized intelligences, [96]’s ([96]) triarchic theory of intelligence, Gardener’s multiple intelligences ([29]; [71]), and others. That said, a task force of the American Psychological Association ([76]), in a very detailed study on defining intelligence, stated that “individuals differ from one another in their ability to understand complex ideas, to adapt effectively to the environment, to learn from experience, to engage in various forms of reasoning, to overcome obstacles by taking thought” (p. 77).

Because this is a brief survey of the past and present literature on intelligence in relation to moral development, we will not go into depth regarding the limitations of each theory mentioned above.

### 2.2. A Narrower Conception of Intelligence: The “Starter Kit”

One issue, though, seems to be common among all perspectives, because there appears to be agreement that there are both genetic and environmental components underlying intelligence ([28]). The issue is the degree to which genetics influences one’s intelligence. It is beyond the scope of this article to delve into detail about this controversy, but we would like to call attention to two recent approaches that are both thought-provoking and grounded in a great deal of cross-disciplinary empirical research.

The first approach is the biological basis of intelligence, more specifically brain development. Research on brain development is more available now because of the much-improved ways we can investigate the physiological aspects of intelligence. [97] ([97]) reported that this research has involved areas such as brain volume, neural efficiency, and performance in different regions of the brain ([15]). One helpful introduction to this area of study is [51]’s ([51]) target article in *Behavioral and Brain Sciences*. As Jung and Haier write:

We propose a model—the *Parietal-Frontal Integration Theory*, or *PFIT*—that elucidates the critical interaction between association cortices within parietal and frontal brain regions which, when effectively linked by white matter structures, …underpins individual differences in reasoning competence in humans, and perhaps in other mammalian species as well.([51])

Reading Jung and Haier’s target article together with the accompanying commentaries gives a nicely focused survey of the literature of the time. However, [97] ([97]) noted that most of these studies have typically produced correlational data, not cause-and-effect conclusions, which are difficult to achieve.

In a similar line of research on the neurobiological foundation of human intelligence, results have shown a relationship with genetics. Several questions arise from this research, such as (a) are people with bigger brains smarter, (b) which brain areas are important for intelligence, (c) does age matter, and (d) are there cell interactions that influence intelligence ([37])?

The second approach, which we will describe more in depth, is that of cultural evolutionary psychology as expounded by [46] ([46]) in her work about “cognitive gadgets.” She states that:

We human beings have created not just physical machines … but also mental machines; mechanisms of thought embodied in our nervous system … These distinctively human cognitive mechanisms include causal understanding, episodic memory, imitation, mindreading, normative thinking, and many more. They are ‘gadgets’ rather than ‘instincts’ … because, like many physical devices, they are products of cultural rather than genetic evolution (pp. 1–2). [Please note the reference to *normative thinking* in her description.]

As she further explains it: “At the heart of the cognitive gadgets theory, of cultural evolutionary psychology, is the idea that social interaction in infancy and childhood produces new cognitive mechanisms, it changes the way we think” ([46]). But, while her view plainly stresses the environmental component of the development of intelligence, Heyes recognizes the need for what she calls a “starter kit,” and here is her outline of what that involves:

The genetically inherited differences between our minds and those of our ancestors are small but very important. They enable the development of Big Special cognitive mechanisms in three ways. First, genetically inherited changes in *temperament* helped to make us remarkably social primates… Second, genetically inherited *attentional biases* ensure that the attention of human infants is locked-on to other agents from birth. We are driven from early infancy to look at biological motion and faces, to listen to human voices… Finally, humans have uniquely powerful *central processors*: mechanisms of learning, memory, and control that extract, filter, store and use information.(p. 53)

Having identified these three pieces of the “starter kit” as fundamental, Heyes continues by characterizing how these mechanisms operate in specifically human cognitive psychology. That is:

Each of these processors is domain-general, crunching data from all input domains using the same set of computations, and taxon-general, present in a wide range of animal species. However, humans genetically inherit central processors with unprecedented speed and capacity. Shaped and fed throughout development by the torrents of information flowing in from other agents, domain-general processors not only capture this information but use it to build new, domain-specific cognitive mechanisms—the Big Special mechanisms that make human such peculiar animals. Thus, the Big Special mechanisms are designed by cultural evolution, but they are built in the course of development by souped-up genetically inherited mechanisms of learning and memory, using raw materials that are, from birth, channeled into infant minds by genetically inherited temperamental and attentional biases.(pp. 53–54)

If this account is anywhere near being correct, one can see how there could be genetically induced variation in the functioning of these “starter kit” cognitive mechanisms, even though the main action is in the multiple ways social interactions help to produce our supply of cognitive gadgets. This could give us a place for a relatively narrow conception of intelligence as something akin to what is measured by Spearman’s *g* (general) factor.

To anticipate, how might the foregoing account of intelligence be connected specifically to moral development? One interesting suggestion is found in the work of Andrew Shtulman. [90] ([90]) compactly surveys a wide variety of evidence about how various imagination-expanding factors may work across a variety of domains such as history, science, religion, mathematics, and ethics. He looks at the way examples, principles, and models can expand children’s developing imaginations. Here is part of his take on the role of learning principles in relation to ethics:

Moral principles, like causal principles and mathematical principles, must be learned … Our moral imagination is initially characterized by parochiality and self-interest and may remain so if not for moral instruction … we will explore the development of three principled distinctions in moral imagination: [1] that bad outcomes are not always caused by wrongdoing, [2] that the way things are is not always the way they ought to be, and [3] that an equal distribution of resources is not always an equitable distribution. These distinctions are a far cry from the higher-order principles that define modern ethics … but *they are prerequisites for learning such principles*.(pp. 147–48)

### 2.3. A Broader Conception of Intelligence: Mindware

That said, many prefer a broader conception of intelligence. Thus, [28] ([28]) presented a definition of intelligence as “the capacity to understand the world, think with rationality, and use resources effectively when faced with challenges” (p. 231). And we can add to this definition that it “enables us to guess better, and the unexpected orderliness is the chief means of doing this” ([5]). Barlow maintains that “intelligence manifests itself chiefly by leading to the right instead of the wrong answer” (p. 208). *We suggest that processes that enhance the likelihood of getting better answers to questions would then contribute to the development of our intelligence*, *and*, *insofar as those processes contribute to getting better answers on the domain of ethics and morality*, *they would contribute to moral development.*

To deepen our discussion of a broader concept of intelligence we will visit the conception outlined by [95] ([95]), starting with his book *What Intelligence Tests Miss: The Psychology of Rational Thought*. Stanovich argues in favor of reserving the term “intelligence” for “the mental abilities measured by intelligence tests” (pp. 12–15). In the outline we are developing, such abilities would be linked to [46]’ ([46]) “starter kit” of cognitive processes described above, with the necessary caveat that intelligence tests’ measurements themselves often reflect how those cognitive processes have enabled the grasp of several sorts of items commonly tested for.

Stanovich is inspired by the work of Amos Tversky and [52] ([52]) (see Kahneman’s *Thinking, Fast and Slow* 2013) to create a dual process picture of how we think, with one type of processes that are the “autonomous mind” and a second type of processes that are the “algorithmic mind.” The autonomous, or Type 1, processes like facial recognition are relatively automatic and fast and demand less energy. In contrast, the algorithmic, or Type 2, processes are relatively slow and computationally expensive, tend to operate in serial rather than in parallel fashion, and are what is happening when we are engaged in conscious problem-solving efforts. Furthermore, as [95] ([95]) sees it, “one of the most critical functions of Type 2 processing is to override Type 1 processing” (p. 22). Finally, in his picture of how human thinking works, Stanovich goes beyond Tversky and Kahneman ([52]) to posit a third set of processes which he calls the “reflective mind.” We will return to this third part of what is now a tripartite model in the concluding part of this paper, but for now we will be attending just to the first two parts of the model.

At this juncture, our readers should be aware that several cognitive neuroscientists have questioned the distinction between Type 1 and Type 2 processes (See the target article “Advancing Theorizing About Fast-and-Slow Thinking” by [21] ([21]), the accompanying commentaries, and then De Neys’ response to the commentaries in a recent issue of *Behavioral and Brain Sciences*). One issue is whether the distinction is not so much a matter of speed, as whether the processes involved are “model-based” or “model-free” algorithms for learning and decision-making. According to [40] ([40]):

Model-based judgment [corresponding to Type 2 systems thinking] is based on an explicit representation of cause-effect relationships between actions and outcomes, plus values attached to outcomes. Model-free judgment [corresponding to Type 1 systems thinking] depends, instead, on accessing values attached directly to actions based on prior reinforcement. (p. 35)

Nevertheless, in considering the relation between intelligence and moral development, [95]’s ([95]) model allows that our Type 2 processes can employ courses of action that are learnable, like the three foundational moral principles that are learned at an early age, according to [90] ([90]). And we can go on learning relevant “mindware,” using the term Stanovich adopts from [79] ([79]), including mindware that is relevant to moral decision-making.

But mindware can be missing or be flawed in its operation, and in either case that will hinder making good decisions. An important illustration of missing mindware in the area of thinking about probabilities, one which draws on Tversky and Kahneman’s ([52]) work, can be shown by asking about the probative power of a particular medical test. In this example a person is asked to presuppose that the test for a (made-up) medical condition, hepatitis Z, is highly *sensitive*, with a 99% rate of delivering a true positive result and a corresponding false negative rate of 1% when someone has the disease. The test is also highly *specific*, with a 95% rate of delivering a true negative result and only a 5% false positive rate, when someone does not have the disease.

Kahneman and Tversky tested people with variations of this scenario: if we suppose a patient, Henry, we will call him, tests positive for hepatitis Z using the test described above, what are the chances that he does in fact have the disease? The most common response selected by people whom they asked was that, given his positive test result, Henry was highly likely to have the disease. However, it turns out, the better answer is that *we cannot tell without some highly relevant specific information,* namely what is the base rate of the infection in the population that Henry is part of. So, if the base rate is, let us say, that 2/10,000 in that population have the disease, then the problem is that, while our test with a high true positive rate may very well catch the two persons who have the disease, given the false positive rate of 5%, then 5% of the remaining 9998 who do not have the disease, that is around 500 people, will also test positive. Therefore, if Henry is a randomly tested person from such a population, his chance with a positive test result of actually having the disease is only about 1/500—far, far from highly probable—and thus the result is very likely to be a false positive. This basic application of Bayes’ theorem is now a common part of medical education, so physicians are much more likely than in the past to have a relevant piece of mindware installed by their education.

Can there be relevant mindware for moral decision making? It would certainly seem so, starting with something like the three principles reported by [90] ([90]), but going on to encompass a wide variety of ethical principles, learned both from modeling and by explicit instruction. So, any assessment of the relation of intelligence to moral development will need to look at factors that enhance or hinder the likelihood that these pieces of mindware will be taken on board. We will return to this topic later with some specific suggestions, but for now here is a story that illustrates what mindware to aid or to hinder ethical decision making can look like.

In his book, *Moral Development and Reality*, 4th ed., [33] ([33]) tells us about a young man named “Mac” in a juvenile corrections center who, it appears, is suffering, not so much simply from absent mindware, but from the active presence of flawed mindware. Mac had resisted and yelled profanities at a staff member who, following institutional policy, was trying to inspect a bag he was carrying. In a meeting with a mutual help group of Mac’s peers and a staff member, the peer group sought to identify the thinking errors underlying Mac’s behavior:

Mac explained that the bag contained something very special and irreplaceable—photographs of his grandmother—and that he was not going to let anyone take the photos from him. Mac’s peers understood his point of view but saw it as one-sided: Mac thought only of safeguarding his photos, without considering for a moment the staff member’s perspective or the facility’s necessary rules… [The staff member] was only carrying out institutional policy concerning possible contraband. Nor did Mac consider that she was not abusive and that he had no reason to assume that the photos would be confiscated. Generating the anger and the overt behavior identified as an Authority Problem, then, were “Self-Centered” and “Assuming the Worst” thinking errors. (p. 203)

One thing to note regarding these and other mindware issues is that they are often not apparent to the person suffering them. Indeed, one of the functions of the peer support group in the story of Mac is to help Mac realize that he has these erroneous thinking patterns so that he can begin to repair or replace them. So, if well-functioning mindware can lead us to make better, more explainable decisions both in general and in our ethical decision making in particular, this leads us to think, as the subtitle of [79] ([79]) has it, about *learnable intelligence*.

## 3. Intelligence and Moral Development: Multiple Perspectives and Classic Research

### 3.1. Perspectives on Moral Development

We will not assume that readers of this journal are familiar with the literature on moral development, so we thought it would be appropriate to discuss some of the issues involved in defining this concept. As with defining intelligence, there are also arguments regarding issues surrounding defining moral development. But for the interested reader, we recommend a very informative overview of the whole subject, [33]’ ([33]) *Moral Development and Reality*, 4th edition. Also, for an historical account of the social development of morality, the story told by Hanno Sauer in *The Invention of Good and Evil* ([88]) provides an inventive overview from prehistory to the present.

### 3.2. A Psychological Perspective

From a psychological perspective, one issue that arises involves which theory best defines moral development. For the purposes of this paper, we will use the terms moral development and morality synonymously, given that many elements are similar in both concepts, although the definitions of both terms are different. Similarly, we will use the terms morals and ethics synonymously. Some emphasize moral reasoning (e.g., [19]; [60], [61]; [80]), which includes moral judgment, some emphasize moral behavior (e.g., [4]), via, for example, operant conditioning and/or social cognitive actions, and some emphasize moral affect (e.g., [24]; [34]). As with the discussion on intelligence, it is beyond the scope of this article to discuss this issue in detail (See [27] ([27]) for various aspects of moral development).

Philosophers, such as [54] ([54]), have defined morality as “prescriptive norms regarding how individuals ought to treat one another” (cited in [57] ([57])). One part of morality is the moral rules. According to [94] ([94]) and [107] ([107]), aspects of moral rules include “generalizability, impersonality, obligatoriness, and independence from authority jurisdiction and punishment mandates, which differ from social-conventional rules and personal jurisdiction” (cited in [56]). And according to [44] ([44]) and [110] ([110]), research on children through adults, from both rural and urban settings, from various cultures and socioeconomic statuses (SESs), and from various religious backgrounds share a common view that rules about fairness and others’ welfare are generalizable, not a matter of consensus, not under authority jurisdiction, and necessary to uphold. (More from this perspective is presented later in this paper. See, for example, ([31])).

### 3.3. A Social Perspective

From a social perspective, our moral rules or principles are developed for use by societies and sometimes aimed more at specific communities. [109] ([109]) state that, according to [106] ([106]) “[m]oral awareness leads humans…to think critically about what rules, customs, and policies that are put in place will likely advance a shared vision. Getting this right is a difficult intellectual challenge” (p. 36). Rules place restrictions on you and add to your obligations that you should obey, but as Wagner et al. point out, “sometimes you will be asked to help challenge these duties and obligations” (p. 37).

### 3.4. A Neuroscience Perspective

With regard to neuroscience and moral development, a milestone in the literature about research on neuroscience and moral development is the comprehensive research review by [75] ([75]). They “hope to demonstrate to readers the potential usefulness and applications of neurobiological and neuroscientific work,” while at the same time counseling against a reductionistic approach in favor of “a holistic approach toward explaining human moral behavior” (p. 290). The research they cite relates to two claims: (1) “healthy moral functioning requires proper brain functioning,” and (2) “brain studies corroborate some, but count against other, traditional concepts of moral functioning” (p. 290).

They report that, with the facilitation of technology such as functional magnetic resonance imaging (fMRI), neuroscience has contributed to the study of moral development. One item they emphasize is the importance of the prefrontal cortex (PFC) in moral reasoning and moral motivation and/or focus. In support of this, they cite a study by [3] ([3]), who examined

children whose prefrontal cortex had been damaged before the age of 16 months. The damage left them unable to acquire social conventions and moral rules, throughout life…Although normal in language and *intelligence* [emphasis added], these patients exhibit behaviour perceived as antisocial, such as shoplifting, sexually aggressive behaviour and non-responsiveness to punishment. (p. 295)

Other areas besides moral judgment where neuroscience has contributed are the study of moral sensitivity (which includes moral perception, moral imagination, and empathy), moral action (which depends on the PFC), and attachment, which [75] ([75]) refer to as the “foundational phase of infant brain development” (p. 301). Further, Narvaez and Vaydich report that “[n]ot only is there hardwiring for sensitivity, early experience is critical for building a brain that can become morally flexible and responsive” (p. 297). To support this statement, they cite research by [64] ([64]), who found that “[d]evelopmental studies indicate that experience methodically rewires the brain and the nature of what it *has* seen dictates what it *can* see” (p. 297).

As mentioned previously, brain development affects moral development. For example, [75] ([75]) report that

[a]t the age of three, synaptic density reaches its lifetime peak and is 50% greater than in the adult brain. Brain areas develop at different rates and peak periods are established at different times. For example, the prefrontal cortex, vital for moral functioning, accelerates at 8 months of age and reaches maximal density at the age of two, at which point cortical development plateaus until early adolescence and is not complete till nearly the age of 30. (p. 301)

See the rest of Narvaez and Vaydich 2008 for an extensive discussion of the neuroscientific contributions to the study of moral development.

Another influential area of neuroscience research on morality is moral intelligence. [74] ([74]) points out that it is based on emotion, which includes the influence of “caregivers … [and also] cognition and moral character” (p. 77). According to [17] ([17]), moral intelligence may be influenced by experiences early in life (cited in [74]). Moreover, these early experiences are important for brain development. By extension, then, this brain development influences one’s moral intelligence.

However, [57] ([57]) sounded a cautionary note. They applaud the attention that neuroscientific research gave to determining parts of the brain that are used in responding to moral dilemmas. But they are perplexed also, and here is how they explain their perplexity:

It is perplexing because the way that morality has been defined, operationalized, and measured in most neuroscience experiments is wildly discrepant from the extensive development and psychological research on how individuals use moral criteria, evaluate morally charged issues, and make moral judgments. In most neuroscience studies, for example, morality is not explicitly defined; it is taken for granted in the nature of the task. (pp. 241–42)

Fortunately, there is now more attention being given to those matters. One example is the work of [39] ([39]), a psychologist and neuroscientist, who develops an account of morality as based fundamentally on lessening the infliction of harms and who uses that account both to guide his research and to analyze the work of others.

For an accessible survey and commentary on newer research, a place to start is with a chapter by May, Workman, Haas, and Han in *Neuroscience and Philosophy*, [20] ([20]). May and his colleagues follow *three threads* in the research, and the first thread involves the neuroscience of “gut feelings” that are involved in our moral judgments. One form of this research involved studying negative responses to hypothetical situations where people were often unable to articulate appropriate explanations for their responses. This tended to support the idea that moral attitudes and decisions are generated predominantly by automatic gut feelings, whereas reasoning is largely *post hoc* rationalization. May et al. cite some limitations of this theory, claiming for example that the centrality of gut feelings is insufficiently corroborated.

Then they move to consideration of a second thread, moral reasoning. In the reasoning thread, the researcher can pose scenarios to participants and then use brain imaging technologies like fMRI to see what factors influence the particular moral judgment made without having to rely on participants’ articulation of their reasons. After surveying a number of studies, [69] ([69]) argue that

A central lesson is that moral cognition is not localized to one brain area … Instead, our capacity for moral judgments involves a spatially distributed network of areas with domain-general psychological functions that are also relevant to moral evaluation, such as understanding the consequences of an agent’s actions, the agent’s mental state, how the action was causally brought about, and the social norms it violates. (p. 25)

Lastly, the third thread considers learning in general and moral learning in particular, and a lesson they draw from the studies in this thread is that we use domain-general decision-making systems which are also used to make specifically moral judgments.

[69] ([69]) conclude their review by touching on the study of moral development where there has been research which compares and contrasts the brain activity of adults with that of children at various ages. One interesting perspective is suggested by studies which seem to show that brain regions that support thinking about self are involved in moral reasoning. These studies lead them to favor an integrative perspective when they cite [6]’s ([6]) account of [82]’s ([82]) model, wherein moral development rests on the cooperation of four components: (1) moral reasoning, (2) moral motivation, (3) character, and (4) sensitivity.

One final note is that the interested reader can find a convenient source of a variety of views on this general topic in [68] ([68]), which not only presents in *Behavioral and Brain Sciences* a précis of May’s book *Regard for Reason in the Moral Mind*, but also includes numerous commentaries on his views and a response to the commentaries from May.

### 3.5. Some Recurring Issues

It has been pointed out that, in the moral development literature, which includes reasoning, emotions, and behavior, that there has been a dearth of research on the relation between intelligence and moral development, and there have been contradictory results as well (e.g., [58]
[58], [59]). Apparently, more research has been conducted on the effects of intelligence on children’s moral development, and less on the effects of intelligence on adults’ moral development ([27]). For example, [7] ([7]) suggest that “intelligence can be assumed to affect moral development” (p. 2). (We find the term “assume” troubling.) To substantiate their postulation that intelligence affects moral development, they cited [53] ([53]), who suggested that high intelligence is related to “efficient information processing, [and] more intelligent people should be better able to integrate…information efficiently and make more sophisticated moral judgments and justifications” (p. 2).

Research conducted by, for example, [22] ([22]) and [48] ([48]) with gifted adolescents seems to support these claims. However, it must be noted that there is a critical flaw in these studies; that is, these researchers appear to equate giftedness with intelligence. Their results suggest that gifted students demonstrated better moral reasoning compared to non-gifted students, but the question arises as to how giftedness is assessed and what specifically it identifies. For example, is it cognitive ability, musical ability, or what? In fact, around 30 years ago, [85] ([85]) noted that frequently there is a lack of exactitude in how giftedness is described. This lack is one of the issues we emphasize in this article.

### 3.6. Noteworthy Research Studies from an Earlier Era Until Recent Times

What follows are discussions of some, to us and hopefully to the reader, important and relevant studies on the relationship between intelligence and moral development. In this process we have “jumped through time” with the research discussed. Much of the research is described in detail to demonstrate the methodological problems that existed in the research and how the methodologies have changed over time. The purpose of this paper is *not* to present an extensive report of the research on this topic, but to present literature that may start the discussion of the relevance of intelligence to moral development.

Going back in time, research was done on the relationship of intelligence and moral development over 80 years ago by [113] ([113]). We should be aware that, at that time, research presented in professional journals was often not as rigorous as it is now. Wiggam claimed that “science has established its highly significant and heartening fact that as man evolves in intelligence the higher he becomes in moral character” (p. 261). Without going into great detail regarding his studies, there are several issues that concern us. For example, Wiggam bases too much of his research on the findings of Terman’s development of the Stanford-Binet intelligence test. Students with scores of 140 and higher were considered highly intelligent. Wiggam continues that Terman then gave his sample of children “numerous special tests which are as reliable for testing moral character and habits as intelligence tests…” (p. 262). Unfortunately, there is no mention of what these tests were. However, he does mention that the tests were good for measuring honesty, willpower, and self-control, which he notes are key moral features. The idea that self-control is a key feature of morals is supported by the delay-of-gratification research conducted some 30 years later by [70] ([70]).

Additionally, [113] ([113]) found that children of higher intelligence were less dishonest than children of lower intelligence. But, unfortunately, no reliability or validity data are reported on these tests of morals.

[113] ([113]) also reported research by [43] ([43]), who gave tests of moral behavior and judgment, as well as intelligence (based on Terman’s work), to over 10,000 students aged 8 to 16 to assess the propensity of these children to lie, cheat, and steal. Wiggam reports that the children with higher intelligence were “more honest than the students with lower intelligence” (p. 264), as measured, presumably, by the Standford–Binet intelligence test.

Moreover, Wiggam states that “[b]rilliant children tend to choose the right conduct simply because they see it is the course of action that promises the best consequences” (p. 263). Based on Terman’s research, [113] ([113]) reported that gifted and non-gifted girls scored higher on the moral tests than did gifted and non-gifted boys, who had a decrease in scores after age 18. Again, he equated intelligence with giftedness. He explains the drop in moral scores in boys stating this “shows … that when boys reach about 18 they have a harder time adjusting themselves” (p. 263).

About 15 years later, [91] ([91]) used the Moral Judgment Interview (MJI) to assess four parts of moral judgment: (1) Objective Responsibility (r = 0.86), (2) Immanent Justice (r = 0.82), (3) Solutions to Transgressions (r = 0.88), and (4) Meaning of Rules (r = 0.72), which indicates that the measure was reliable. The results show that students of high intelligence, as compared to students of average intelligence, had higher scores for Objective Responsibility and Meaning of Rules. *Interestingly, older students of average intelligence scored higher than younger students of high intelligence in these two areas.* Another interesting finding was that girls had higher scores for Solutions to Transgressions, with one exception. That is, 7- to 8-year-old boys of average intelligence scored higher than girls of average intelligence on this variable.

Furthermore, research by [47] ([47]), in relation to [80]’s ([80]) stages, studied how intelligence affects children’s moral development. His participants consisted of 100 elementary, middle-class, boy and girl students from a Jewish parochial school in a large city in the east of the United States. Obviously, this is a potentially biased convenience sample, and thus the results should be interpreted with caution. Children’s intelligence was measured using the Stanford–Binet Intelligence Test, Form L-M. Twenty-five children of average intelligence and twenty-five of high intelligence were randomly selected from 7- to 8-year-old second graders, and from 11- to 12-year-old third graders. The range of IQ scores for average students was 91 to 109 (mean = 103) and 124 to 156 (mean = 136) for brighter students. [47] ([47]) stated that

[o]ne could reasonably expect more intelligent children to be less concrete and more flexible in their thinking, possess a higher level of conceptual ability, and have greater insight into the social processes than less intelligent children, and that these factors should have a significant impact on their perceptions and evaluations of moral issues. (p. 28)

In a later, related study, [102] ([102]) conducted a review of the literature available at that time on moral sensitivity and moral judgment in relation to giftedness. One will notice that she appears to equate giftedness with intelligence. In her account, moral sensitivity includes

reading and expressing emotiontaking the perspective of otherscaring by connecting to othersworking with interpersonal and group differencespreventing social biasgenerating interpretations and optionsidentifying the consequences of actions and options ([73]).

In her review, [102] ([102]) reports the finding by [72] ([72]) that, although high-academic ability (HAA) students tend to score higher on the Defining Issues Test (DIT, [82]) than average academic ability (AAA) and below average academic ability (BAA) students, it does not consistently predict scores in moral judgment. Moral judgment is frequently assessed by the DIT, which is based on the moral reasoning stages of [61] ([61]). One issue to consider is that academic ability is not equal to intelligence. Academic ability is related to intellectual ability (e.g., knowledge, logic), but it is not the same as intelligence (e.g., astuteness, cleverness). Another issue in [72]’s ([72]) study that may have influenced the results was the use of middle-school students from a private preparatory school and from a public school, both from suburban areas in a mid-Western state. She mentions this issue briefly by stating “[a]s is evident from the overall means…this appears to be a highly selected sample” (p. 273).

In another line of research, [74] ([74]) studied moral functioning, where she stated that “[e]motions form the foundations of brain functioning in terms of motivation and intelligence, but these are not enough for mature moral functioning” (p. 83). Previously, Narvaez noted that early life experiences in developmentally appropriate environments influenced brain development. In addition, these environments should allow for “deliberative” practice. According to Narvaez, these environments rely on the “conceptual structures that derive from experience” (p. 83).

[102] ([102]) also cites research by [55] ([55]), who assessed (using the DIT) the moral development of gifted students and their intelligence with the Wechsler Intelligence Scale for Children (WISC-R, [111]). [55]’s ([55]) results showed that, based on moral reasoning/judgment and scores on the WISC-R, gifted students “appeared to reach a relatively high stage of moral reasoning earlier than their chronological peers” ([102]). But, as mentioned earlier, the results indicate that this finding is not necessarily the case with adolescents ([72]).

Of interest, though, to [102] ([102]) is research on the moral sensitivity, which is one of the components of [83]’s ([83]) Four Component Model, of Finnish gifted students and Finnish Academic Olympians (AO). (In Finland, AOs focus on math, physics, and chemistry. These are individual competitions; there are also team competitions, such as in the United States.) Again, we have concerns with the apparent equating of giftedness with intelligence, as well as being an Academic Olympian with giftedness. For example, is an AO gifted or of high intelligence? By gifted, is the reference only to cognitive giftedness? Is being an AO due to academic achievement, or to other variables?

That said, [102] ([102]) notes that “*high intellectual ability* [authors’ emphasis] does not predict mature moral judgment. Furthermore, responsible moral judgments for the moral dilemmas in science [reported in this review, i.e., ([102])] require moral motivation and moral sensitivity” (p. 63). Focusing on future scientists, she states that they need ethical (moral) knowledge that also includes skills in “ethical sensitivity, ethical judgment, ethical motivation, and ethical action” (p. 63). She suggests, then, that education should support not just students’ cognitive capacity, but also their social and emotional (affective) capacities.

As our journey progresses, there have been some interesting findings relevant to those presented above. For example, [7] ([7]) studied gifted and non-gifted elementary students, and they found that moral development was *not related* to intelligence. Because of this, we felt it necessary to delve into Beißert and Hasselhorn’s study in more detail.

However, there are issues regarding their study that may have influenced the results. For example, there were 129 participants: 62 from regular primary schools, and 67 from enrichment programs for gifted children from southwestern Germany. (Students were assigned to gifted programs based on nominations by teachers. More than one program and school were included, because they wanted to achieve a certain number of intelligent students. How intelligence is delineated is noted below.) The age ranges were 6 years and 4 months to 8 years and 10 months, with no significant differences in age and gender between both groups.

In addition, students were interviewed individually either in separate rooms in their schools or in their homes by trained research assistants; inter-rater reliability was not mentioned. They did, however, audiotape and later transcribe the tapes. Again, no reliability data were reported. But why did they interview students in different environments, which may have affected their results? For example, the children obviously would be more familiar with their homes, and perhaps there were distractions that could not be controlled in that environment.

Intelligence was measured with subscales 3 (classifications), 4 (similarities), and 5 (matrices) of the Culture Fair Test 1 (CFT1; [11]). [7] ([7]), citing [112] ([112]), stated that “[a]ll three subtests have in common that they focus on relational thinking and the comprehension of rules and regularities which can be understood as a central part of general intelligence” (p. 4). Intelligence scores ranged from 82 to 145 (M = 112.4 for the regular primary school students and M = 122.3 for the gifted program students). [7] ([7]) caution that the norms were based on studies in the 1990s.

Beißert and Hasselhorn used four picture stories depicting different “moral transgressions” to measure students’ moral development. One story was about “not sharing with a needy child,” another about “stealing another child’s candy,” a third about “hiding someone’s property,” and a fourth about “picking on someone” (p. 4). According to Beißert and Hasselhorn, “[t]his type of moral transgression stories has been frequently used and has been shown to be valid” (e.g., [23]) (p. 4).

Students’ reasons for their responses to the various stories were coded into four categories: (1) moral reasons, (2) sanction-oriented, (3) hedonistic, and (4) undifferentiated (unelaborated or uncodable) (p. 4). Coding of interviews was conducted by Beißert, and approximately 25% of them were coded by another researcher, with high inter-rater reliability (Cohen’s *k* = 0.89).

Two indices were computed: Strength of Moral Motivation (SMM; [65]) and Negatively Valenced Moral Emotions (NVME; [77]), such as feeling guilty. These indices “combine attributed emotions with the corresponding reasoning” ([7]). According to [7] ([7]), “both–attributed emotions and justifications–help us understand children’s (moral) motivation” (p. 5). High scores on SMM signified strong moral motivation, and high scores on NVME signified strong negative emotions (e.g., very guilty) for the stories.

Their results were contrary to most of the studies discussed previously. For example, they found no correlation between intelligence and moral reasoning. Moral reasoning was assessed by analyzing justification responses to the four moral transgression stories: picking on someone, stealing, hiding someone’s property, and not sharing. Combined, these four stories represented moral motivation. Subsequent analyses showed no significant effects of intelligence using any of the four stories mentioned previously. Beißert and Hasselhorn also found no correlation between intelligence and attributed emotions (i.e., very bad, rather bad, rather good, or very good) using any of the four stories. Further analyses indicated that intelligence was not a significant predictor of either moral motivation or NVME (e.g., guilt) (See [7] ([7]) for detailed data regarding their statistical analyses). Lastly, they found no correlation between moral development and intelligence using any of the four stories, nor was there a correlation between intelligence and moral motivation or the NVME index.

In sum, Beißert and Hasselhorn state that, for children aged between 6 years, 4 months and 8 years, 10 months, “inductive reasoning competencies [i.e., what they consider intelligence] cannot explain differences in moral development” (p. 7). This suggests that findings from prior research with adolescents or adults cannot simply be extended to younger participants [children]” (p. 7). To us, a problem with their conclusion is that inductive reasoning ability is not the only component of intelligence.

As pointed out, though, by Beißert and Hasselhorn, the differences in their results as opposed to previous research may be due to several factors, such as:Their technique used more realistic moral scenarios, as opposed to those used in the Moral Judgment Interview and the DIT; thus, responses may not truly measure one’s moral development/morality. Note, the use of non-realistic dilemmas has been criticized in the past. See, for example, Slavin’s *Educational Psychology*: *Theory and Practice*, 12th ed. ([93]).Many previous studies involved gifted students, but how these students were identified as gifted and what determined their giftedness (e.g., cognitive ability) are unknown.Their method was similar to that used in many studies on prosocial behavior that found, for example, no relationship between intelligence and working memory and sharing behavior, which is part of prosocial behavior, which is considered part of moral development (See for example, [78]).

Several years later, [87] ([87]) stated that “research on the regulation of ethical sensitivity in persons with HIA [high intellectual ability] is scarce and necessary, [that] suggest[s] that children and adolescents with HIA are superior in ethical sensitivity than their typical peers” (p. 1). Because of this lack of research on ethical sensitivity, they conducted a study of 44 students aged 8 to 10 years old to determine and to compare the ethical sensitivity of students with HIA (N = 21) who attended the Extracurricular Enrichment Program at a university in Spain to an age-matched group of students of average intelligence, who attended a public school (N = 23). Surprisingly, intellectual ability/intelligence was not measured by a standardized intelligence test (e.g., the Stanford–Binet, WISC, or WAIS-R), but by a professional’s judgment. To assess ethical sensitivity, [87] ([87]) used the valid (alpha = 0.93) Spanish version of the Attitudes Toward Human Rights Inventory (ATHRI; [10]) that was used to assess three factors of attitudes toward human rights and civil liberties. These factors are (1) personal liberties, (2) civilian constraint, and (3) social security (e.g., government providing welfare services).

[87] ([87]) reported results that seem to show that students with HIA have greater ethical sensitivity than do students with “typical” intellectual ability. (Are they referring to “average”-intelligence students when they refer to them as “typical”?) [87] ([87]) speculate that this is most likely assisted by the intricacy of these students’ intellectual abilities. The results support the findings of, for example, [2] ([2]) and [84] ([84]). Additionally, they found that ethical sensitivity developed earlier in HIA students. That is, it developed in 8- to 9-year-olds, but decreased in 10-year-olds. Perhaps, as they speculated, children become more pessimistic as they age, which supports findings by, for example, [81] ([81]) and [108] ([108]).

However, there are several serious problems with the study. First, they used a small convenience sample, and they failed to specify the number of boy and girl students. Second, the “professional” who assessed students with HIA is not specified. That is, what kind of professional was s/he? Furthermore, there was no inter-rater reliability reported if this “professional” used a measure (e.g., an observation checklist) to assess HIA.

[87] ([87]) state that there is still a dearth of research on the differences in ethical sensitivity of 8- to 10-year-old students with HIA to students of “typical” (average) intellectual ability. We add that further research with children, adolescents, and young adults of all education levels, not just children aged 8 to 10, is needed. And, to have greater value, such research should give due attention to issues of sampling and the way in which assessment of intelligence and assessment of moral development is performed. In addition, Sastre-Riba and Camara-Pastor suggest that ethical regulation should be taught to HIA students, but we suggest that ethical regulation is an important skill for students of all intellectual abilities to acquire.

Finally, we agree with [87] ([87]) that the study of how students with different intellectual abilities solve complicated ethical problems to improve our world is important in the 21st century. Some other items that should be addressed in future research are:using participants who are below the average range of intelligence (e.g., [7]). (Note that this brings up again the issue of how intelligence is measured),including preschool children in one’s research (e.g., [7]) (Note that this is suggested previously in this paper), andconducting cross-cultural studies.

## 4. Different Take on Morality

We would like to suggest an avenue for further study that we think has serious potential. First, we will outline the initial depiction of “common morality” by [30] ([30]), which includes a view about goods and evils, a set of moral rules, an argument for them, and a description of a process for justifying, on occasion, the breaking of a particular moral rule. It also includes a distinction between moral rules and moral ideals. If Gert’s depiction of common morality is anywhere near the mark, then learning the basic moral rules is not a difficult challenge for most people with normal intellectual and emotional ability. There are, of course, individual and social circumstances which might lead to people not grasping the rules or not abiding by them, and there can be numerous complications that occur when trying to apply the rules, for example, in ethical decision making in medicine or police work. Our intent here is simply to suggest that assessing moral development need not be confined to instruments that rely on either a Kant-inspired view of morality (see [60]) or some version of utilitarian consequentialism. We wish to motivate interested readers to explore Gert’s perspective with a view to the possibility of creating a valid measure of moral development inspired by it.

### 4.1. Gert’s Depiction of Common Morality 1: Thoughts on Good and Evil

People wonder occasionally about the nature of good and evil, and here is a challenge by Richard Alexander, a sociobiologist:

the view of a human history developed so far in this book suggests that humans behave **as if** they are concerned with their own genetic interests, and they are also masters at deceiving others. I suggest that the separateness of our individual self-interests, and the conflicts among us that derive from this separateness, have created a social milieu in which, paradoxically, the only way we can actually maximize our own self-interest and deceive successfully is by continually denying—at least in certain social areas—that we are doing such things … The result, I believe, is that in our social scenario-building we have evolved to deceive even ourselves about our true motives.([1])

Alexander suggests that we have it all backwards, and that we are “programmed” so that it keeps on looking correct to us even as we ignorantly befuddle ourselves. Is what is being said here that we are simply mistaken about our motives? That when we think we are behaving unselfishly, we are really concerned with our own genetic interests?

But it looks like a mistake is being made at this point that is evidently all too easy to make. The mistake is in going from a theory that attempts to explain the social behavior of human beings and other animals, a theory that is developed in terms of genes competing with one another to be reproduced, to the conclusion that unselfishness and other “noble” motives are just cloaks for the competitive vying of our genes. Even if one grants a great deal to the explanatory power of such a theory, what it serves to do is to explain why, for instance, parents may sacrifice themselves for their children. *But explaining why this sacrifice is made does not make it any less a sacrifice.* To explain something is not necessarily to explain it away as an illusion.

So, let us look at the two basic traditional schools of thought about good and bad: (1) hedonistic theories and (2) what we will call “human potential” theories. Regarding hedonism, a number of years ago the brother of one of the authors announced in a tone of voice that one usually reserves for religious revelations that “[e]xperiences are all that really matter. Nothing matters, nothing is important unless somebody, somewhere, experiences it.” This, we submit, is the heart of hedonism. The point is that, for the hedonist, what is ultimately of value are our experiences, and these experiences come in two basic types: (1) positive ones that are pleasant, enjoyable, satisfying, exciting, etc. and (2) negative ones that are painful, unpleasant, repulsive, disgusting, etc. Experiences of the first type are *the good,* and experiences of the second type are *the bad.* Everything else is called “good” or “bad” fundamentally because of the contribution it can make to our having these experiences. Of course, many things are mixed, neither wholly good nor wholly bad, but that is just to observe that a given thing can have many different effects. Also, it is plain that a particular sort of thing might be good for person “A” and bad for person “B,” because “A” and “B” are affected differently by the same thing. One further thing to notice is that what is good and bad on this view is an objective matter. Whether something causes a person pain or makes that person feel good is a matter of fact. People often can have some idea of whether a person is pleased or pained, and they can often get it right, while they may get it wrong on some occasions. See Richard Brandt’s *A Theory of The Good and The Right* for an overview ([8]).

So why is not everybody a hedonist in their theory of good and evil? Here are a couple of concerns that lead to hesitation. First is the so-called “hedonistic fallacy” or “hedonistic paradox.” This is the idea that, if you make seeking pleasure and avoiding unpleasantness the central goal of your life, then, if you succeed in grasping that which you think will provide you with ultimate satisfaction, not too long afterward you will wind up becoming dissatisfied, like the stock image of the bored rich kid.

Whether that is true or not, there is a second objection. This second objection claims that being pleased, enjoying oneself, and so on are all states of mind that follow upon the fulfillment of some aspect of our human potential. Therefore, the realization of that human potential is what is really the good. There are many human potential theories, and Abraham [67]’s ([67]) view of human beings as having a hierarchy of needs ranging from needs for food and water to a need for meaningful “peak experiences” is one such theory. Traditional Christian ethics also appears to be a theory of this sort, when it stresses as the most central concern in human life the development of loving relations between people. Developing a “human potential” theory is relatively straightforward. You observe someone whom you think really “has it all together,” then you try to figure out what that “it” is. That is *the good*, and anything that thwarts it or stifles it or prevents it from developing is *the bad*.

All of these theories are what might be called theories of the “ultimate good” (or *summum bonum* in the Latin phraseology that is sometimes used). They pick something like pleasure and make it the goal around which the value of everything else is supposed to be determined. [30] ([30]) develops a quite different approach in *The Moral Rules.* To begin, Gert focuses on *evils*. This is interesting if only because, in the history of discussing these issues, much more time has been devoted to talking about the good than to talking about the evil. Even more importantly, “*evils*” is plural, suggesting that there will not be any attempt to find an “ultimate evil” or *summum malum*.

Gert defines as evils all those items that it would be crazy to want for yourself without some special reason. Death, pain, disability, loss of freedom, and loss of opportunity come readily to mind as items that fit his definition. Note carefully that this does not say that it is always crazy to want to die, etc. You might have some special reason, like being in intolerable pain that nothing could stop, that might make wanting to die a reasonable wish. But reflect for a second on why it would be crazy to want these things without some special reason. The point is that they interfere with your efforts to do what you want to do. Needless to say, death interferes drastically with one’s plans, and being ill or confined or in pain generally does not make things easier. So those things are evils which ordinarily are obstacles to our accomplishing our purposes. On the other hand, *goods*, as one might guess, are going to be those things that generally aid us in accomplishing our purposes. Thus, health, intelligence, knowledge, strength, power, wealth, and liberty are examples of goods.

Notice the inversion of perspective. Instead of searching fruitlessly for something that could appropriately be called the *summum bonum*, or the *ultimate* good, we instead have our minds directed toward the *means* for accomplishing our diverse purposes. Now we can speak of what is bad and what is good with an eye toward what serves human purposes. We can speak of things as “bad” with less and less qualification the more likely they are to hinder the accomplishment of human purposes, and we can speak of things as “good” with less and less qualification the greater the likelihood that they will enhance the accomplishment of human purposes. All the while, of course, we acknowledge that what is ordinarily an evil or a bad may be a good for a particular person in unusual circumstances, and likewise that which is ordinarily a good may interfere with someone’s plans and thus be an evil for him or her.

If the foregoing sounds pretty much like common sense, that should recommend it to us. On this understanding, there is no big problem about evils not really being evils and goods not really being goods. Nor need there be any general skepticism about good and evil. Such skepticism may be encouraged by “ultimate good” approaches, since it is hard to identify anything credibly as the one and only ultimate good, but the temptation to skepticism is lessened on Gert’s approach. While it may be quite difficult, perhaps even impossible, to know that a particular thing will be a good or a bad for a specific person, especially if we do not know the person involved, that does not mean that our everyday knowledge is fatally flawed about what people in general avoid because it hinders them and what people in general seek because it helps them. If our knowledge of these matters were fatally flawed, we simply could not live in association with other people. Note at this point that nothing has been said about right and wrong, two sister concepts of good and evil. There is a relationship between the two pairs of concepts, but they are not identical notions. That discussion is for the next section.

### 4.2. Gert’s Depiction of Common Morality 2: The Moral Rules

In Gert’s depiction, *the point of morality is to lessen the occurrence of evils*. Hence, the moral rules prohibit inflicting evils, and, since the moral rules prohibit the inflicting of evils, presumably one would *publicly advocate* that they be followed with regard to oneself and those that one happens to care for.

However, if one is to reach agreement with other rational people, then one must advocate that the rules be obeyed by all people with respect to all other people. [30] ([30]) defines the rational person’s public stance toward each of the moral rules as follows: “Everyone is to obey the rule with regard to everyone else except when he would publicly advocate violating it. Anyone who violates the rule when he would not publicly advocate such a violation may be punished” (p. 96, emphasis added).

Gert continues: “Only those rules toward which all rational men would publicly advocate this attitude count as genuine or basic or justifiable moral rules” (p. 96, emphasis added). Here are some unsurprising moral rules that pass this test:Do not kill.Do not cause pain.Do not disable.Do not deprive of freedom or opportunity.Do not deprive of pleasure. (p. 86)

These rules are very basic in common morality, but there are others Gert thinks can also be justified and are thus worth mentioning:Do not deceive.Keep your promise.Do not cheat.Obey the law.Do your duty. (p. 125)

There is no claim that this is an exhaustive list, so other rules like “do not steal” may be added.

But Gert quickly continues that, since the point of the moral rules is to lessen the occurrence of evils, there can be occasions when breaking a moral rule and inflicting an evil on someone leads to lessening the overall occurrence of evils. Given that circumstance, then there are justifiable violations of the rules, and there can even be unjustified keeping of the rules.

In Gert’s view, the violation must be the sort that one can publicly advocate, and we must consider these relevant factors: (a) the amount of evil the violation will cause and the amount it will prevent, (b) the desires of the person(s) upon whom the evil is going to be inflicted, and (c) the effect that this violation, if publicly allowed, would have on encouraging further violations that are not as justifiable. A consideration that is not relevant to justifying publicly a violation of a moral rule is, according to Gert, if it is done simply in order to obtain some good for anyone for whom one is concerned, including oneself. “Thus, all killing and torturing for pleasure or profit is clearly immoral, whereas, killing and torturing to prevent greater killings and torturing may sometimes be allowed by public reason” ([30]).

One consequence of Gert’s view that there are justifiable violations of the moral rules is visible in developmental research reported by [18] ([18]) who found that “100% of 7–8-year-old children judged it was *wrong* to deceive a teacher who commanded that the student *not* harm others, while 82% of 7–8-year-olds, *positively* endorsed deceiving the teacher when the teacher commanded one student to harm another. This again highlights that deception, while it is a potential tool when needed to address injustice, is not used as a blanket or preferred strategy by most children” ([18]).

Thus, Gert affirms that there will be genuine moral disagreements; that is, rational persons with the best will in the world may disagree about the permissibility of certain violations such as assisting a terminally ill, conscious, and rational patient who wishes to die to do so. However, the disagreements will occur within a larger framework of agreement. For it is very likely that genuine moral disagreements are those that occur within the larger framework, on occasions when it is allowed by reason to publicly advocate either alternative. It is here that each individual has to decide for themselves what violations they would publicly advocate.

The foregoing is like an architect’s sketch of the plan of the ground floor of the edifice of morality. It is important to note that Gert has refined this sketch in more detail, both in response to critics and from his work in applied ethics ([31]; [32]).

At this point we should also note that there is controversy about whether lessening harms inflicted is the foundation of morality. For example, [42] ([42]) and other colleagues developed the view that morality involves operations similar to our having receptors on our tongues for different flavors. Echoing David Hume, [42] ([42]) notes that “[m]oral judgment is a kind of perception, and moral science should begin with a careful study of the moral taste receptors” (p. 136).

As he developed his view of the foundations of moral judgment, Haidt determined that there are five distinct moral taste buds, corresponding to five adaptive challenges faced by our ancestors: (1) care/harm from caring for vulnerable children, (2) fairness/cheating from the need to form partnerships with non-kin, (3) loyalty/betrayal from the need to form coalitions to compete with other coalitions, (4) authority/subversion from negotiating status hierarchies, and (5) sanctity/degradation from keeping oneself and one’s kin free from parasites and pathogens ([42]). He does not claim that this is the complete set, and, indeed, he later adds a sixth, liberty/oppression, to better understand political partisanship.

Haidt’s views are based on empirical findings, but, as with many other theorists and theories, they have not gone unchallenged. One of the most serious challenges is found in the recent work of psychologist and neuroscientist [39] ([39]), whose view is that “every fight about morality comes down to one thing: competing perceptions of harm” (p. 3). Gray develops a detailed critic of the standard interpretations of Haidt’s experimental paradigm, and he challenges the supposition that there is a distinct brain mechanism responsible for each of the moral “tastes.” Gray claims that “[m]odern neuroscience shows that our brains are not divided into different little parts, but instead are composed of interconnected functional networks. These networks operate at very general scales” ([39]). We recommend to the interested reader that they, in effect, put Haidt’s and Gray’s books side-by-side to see how their arguments develop.

And, of course, those competing perceptions of harm that Gray mentions can vary across cultures, as is shown by [92] ([92]), who say:

This chapter spells out the wisdom that ideas about karma, the sacred self, the sacred world, and feudal ethics encode in their metaphors. It applies one of the central assumptions of cultural psychology: indigenous or folk theories (our own and others) should be taken seriously as cognitive objects and as potential sources of social scientific and practical knowledge.(p. 120)

### 4.3. Gert’s Depiction of Common Morality 3: Moral Ideals

Finally, we should note that there is more to Gert’s depiction of the structure of morality. In particular there are, in addition to the moral rules, moral ideals that stress the need for us to act on occasion to bring about some good for certain people. A notable example of acting on a moral ideal would be those physicians who volunteer to work in conflict zones for Doctors Without Borders in order to prevent death and relieve suffering, but we also can often see the ideals present in smaller ways with simple acts of human kindness, like someone extending a helping hand with no likelihood of any return.

Notice that, whereas the moral rules require us to refrain from inflicting evils and thus only require inaction, the moral ideals require positive actions of the kind needed to produce the good effect. So, a part of our moral development is our tendencies to do good, and, while these ideals might not be the foundation of morality that the moral rules are, they are nonetheless very important, and any instrument that attempts to assess our moral development needs to take them into account.

## 5. Intelligence and Morality in Action: Wisdom and Wisdom-Building

In this section we explore the “reflective mind,” the third part of the tripartite structure of the mind as developed by [95] ([95]). We will describe some recent work on wisdom by examining the work of Robert Sternberg and also work on the concept of practical wisdom (*phronesis* in the Greek) that has led to the creation of a validated measurement instrument ([63]). The point is that moral development cannot simply be about a grasp of the relevant rules; there must be a practical wisdom developed that includes noticing the relevant ethical features of situations, being able to envisage a variety of courses of action in response, and the ability to successfully follow the course of action that seems to be the best, morally speaking. This leads to a consideration of what can be broadly called character education, the ensemble of influences that shape the character of the individual for better or for worse.

### 5.1. Sternberg’s Work on Wisdom

The bulk of this section will focus on recent serious efforts to reestablish the concept of practical wisdom, inspired mainly by the ethical works of Aristotle. But before we do that, we would be remiss if we did not call attention to important recent work on wisdom. For example, [41] ([41]) presents an extensive overview of work by wisdom researchers across the globe. One outcome of the overview is identifying a common wisdom model “suggesting that central to the psychometrically oriented operationalizations of wisdom in psychology are moral aspirations (e.g., common good orientation, orientation toward shared humanity)” and what they label as the Perspectival aspects of MetaCognition (PMC), such as “intellectual humility, balance of diverse viewpoints, consideration of diverse perspectives and broader contexts than the issue at hand” (p. 112). In the same spirit, [35]’s ([35]) article “The Wisdom Researchers and the Elephant” develops an “integrative model of wise behavior” that combines both “noncognitive components,” such as an exploratory orientation and concern for others (pp. 358–60), and “cognitive components,” such as life-knowledge, self-knowledge, and consideration of divergent perspectives (pp. 360–62).

Moreover, Robert Sternberg and his colleagues pursued the goal of creating a serious psychology of wisdom in all its aspects. So, in “What is Wisdom: Sketch of a TOP (Tree of Philosophy) Theory” ([98]), he presents a view of wisdom that relates it to the organization of the discipline of philosophy, and he allows that that organization “represents a long-term, trial-and-error learning process that provides a conceptual and taxonomic base for understanding the nature of wisdom” (p. 47). This is an interesting document, although one can quibble a bit with the way he outlines the TOP. For example, he depicts hermeneutics as one of the major branches of the TOP. As he understands it, hermeneutics deals with interpretation, and it crucially involves intellectual integrity, which in turn “requires knowing what is real and what is not, and caring about which is which” (p. 57). This makes it sound like hermeneutics requires answers to the question of “What is real?”, the defining question of the metaphysics branch of the tree, and an answer to the question of “What do we know?”, the defining question of the epistemology branch of the tree. And, while this article provides food for reflection, its taxonomic concern is tangential to our inquiry. Subsequent to the TOP article, Sternberg has another recent piece ([99]), which we address below.

But first, let us consider another one of Sternberg’s recent publications. There is a very useful, relatively short summing up of the main points of work on wisdom in [100]’s ([100]) *Wisdom: The Psychology of Wise Thoughts*, *Words*, *and Deeds*. The book begins with Chapter 1 on “What is Wisdom” and moves to chapters that give accounts of how wisdom has been studied in psychology and how it might be measured, it continues to later chapters on how wisdom develops and how it may be developed, and it ends with a self-assessment instrument in Chapter 8 “Am I Wise?”. We will touch only briefly on what the authors have to say and urge everyone with an interest in the topic to consult the book itself as a start to understanding Sternberg’s views and the motivation for his work.

Wisdom in [100]’s ([100]) view is inherently ethical, because they define wisdom as

the application of one’s world knowledge and skills toward (1) attaining a common good; by (2) balancing one’s own, others’, and larger interests; over the (3) long-term as well as the short-term, through (4) the use of positive ethical values, in order to (5) adapt to, shape, or select environments(pp. 1–2). 

This conveys the flavor of their view, and we will briefly discuss some particular aspects. First, how does wisdom relate to intelligence? Employing the Cattel–Horn ([100]) distinction between fluid and crystallized intelligence, they acknowledge that wisdom relies on each. Some degree of fluid intelligence is needed for the wise person to be able to learn from experience, and wisdom itself is some form of crystallized intelligence. However, [100] ([100]) are quick to point out that, while wisdom may require some degree of intelligence, being wise is not the same thing as being highly intelligent, because highly intelligent people may use their abilities in the service of purely self-interested goals, or they may lapse in a variety of ways in situations that call for wise and judicious thinking. They conclude that “intelligence is no cure or even preventative for foolishness” (p. 152).

Second, in the last chapter they provide an assessment tool to provoke guided reflection on the answer to the chapter title “Am I Wise?”. There are 13 challenging reflection prompts, and for each one the same set of potential responses is provided: (1) never or almost never, (2) usually not, (3) sometimes, (4) often, (5) almost always. To obtain the flavor of the instrument, here are two of the thirteen reflection prompts:

(1) Under “Empathy,” the prompt asks: “When you are faced with a difficult conflict, how often do you try to put yourself in the shoes of everyone involved, even if your immediate feelings clearly place you on one side?” (p. 180).

(2) Under “Openness,” the prompt asks: “Think about how you deal with new points of view and new experiences. How often are you open to doing things in a new way?” (p. 181).

The rest of the prompts are based on other factors that constitute wisdom in their view, but in all the cases the “wiser” answer, as one might suspect, will be (5) almost always.

A third aspect of their view of wisdom is presented in Chapter 6 “How Do We Cultivate Wisdom?”. This chapter has two central foci, the first of which is a discussion of important cognitive fallacies which impede the exercise of wisdom. They discuss eight fallacies, such as the sunk cost fallacy, which occurs when we are confronted with the costly results of a mistaken decision and, instead of cutting our losses, we “double down” and dig into the hole even deeper rather than stop digging. Or another fallacy is confirmation bias, where people mostly tend to seek out information that confirms what they already are prone to believe.

The fallacies are discussed succinctly, but one fallacy that is especially relevant is the fallacy of egocentrism, where we are so caught up with self-centered concerns that we become foolish. As Sternberg and Glück view it, “wise people balance their interests with those of others,” while “foolish people see other people as existing only to serve their own egocentric interests” (p. 131). With this in mind, Sternberg and Glück contend that, to guard against the fallacy, two kinds of thinking are needed: (1) *dialogical thinking,* which means “seeing multiple perspectives on problems” (p. 131), and (2) *dialectical thinking*, which means an awareness that “as perspectives change over time, what people view as correct or appropriate changes as well” (p. 132).

This leads them to address teaching for wisdom. They provide examples of using case studies for class discussions that present problems that require balancing interests and that, at least initially, have no one clear answer. As they say, “[o]ne learns to think wisely by practicing wise thinking” (p. 144). But for them, this practice requires a wise teacher in the mold of Socrates “who helps young people to think clearly without preempting them in deciding what they should think” (p. 144). We shall return to this topic later with some evidence-based suggestions about how this process might take place in a way that does not require a stand-in for Socrates.

### 5.2. Pioneering Work on Rethinking Practical Wisdom

In the past few years there have been several important perspectives on wisdom presented by researchers who stress that we need to consider multiple components when we think about these matters. For example, in [101]’s ([101]) “Moral Intelligence—A Framework for Understanding Moral Competences,” the authors stress that moral intelligence which is “the agent’s capacity to process and manage moral situation” (p. 120) involves as many as five distinct competences: 1. moral compass, 2. moral commitment, 3. moral sensitivity, 4. moral problem solving, and 5. moral resoluteness (p. 122). Among other things, they stress that the self-regulation needed for mobilizing one’s competences is not a matter simply of conscious application of moral standards, but that it “also depends critically on non-conscious, automatic processes” (p. 125).

[99] ([99]) echoes a similar theme in “A Trilogy Theory of Moral Intelligence,” which he defines as “the knowledge, abilities, and attitudes needed to apply universal principles of right and wrong to personal, collective, and societal problems” (p. 1). We suggest that this is very similar to what the tradition stemming from Aristotle, whom Sternberg cites, has called practical wisdom or *phronesis*. This impression is reinforced when we see that he notes that, in addition to considering (1) the processes of moral intelligence and (2) the content of moral intelligence, we must also be seriously concerned with part (3) of the trilogy: “Mediating Forces: Dispositional and Situational Forces that Affect the Translation of Moral Intelligence into Action” (p. 11).

With respect to [99]’s ([99]) work, there are a couple of important issues worth noting. First, when it comes to giving an account of the “universal principles” that are part of the constitution of moral intelligence, Sternberg lists a number of them. They range from the Golden Rule to selections from the 1948 Universal Declaration of Human Rights and to a modern version of the Hippocratic Oath. In explaining his selections, he allows that the moral principles “are based on four kinds of sources: great religious texts, the United Nations Declaration of Human Rights, the Hippocratic Oath, and the moral intelligence of wise leaders” (p. 9). While we agree that those sources present suggestions worth considering, what we do not see is an account of what explains and justifies particular selections. It is at this point that we find helpful [30]’s ([30]) perspective on the moral rules and their basis as publicly justifiable prohibitions on causing harms. The second issue is that Sternberg asserts that “[t]here is, as yet, no fully validated measure of moral intelligence, and it is not clear that there will be one, at least in the foreseeable future” (p. 13). In response, we suggest readers consider recent work on practical wisdom described below, since it has now reached the point where Kristjansson and colleagues claim to have developed such a measure (see [63]).

Another contribution in this area is from [35] ([35]) in “The Wisdom Researchers and the Elephant: An Integrative Model of Wise Behavior.” They cover many different perspectives that have arisen in recent research on wisdom, and they propose their own model, which has as its “core proposition” that “in challenging real-life situations, noncognitive wisdom components (an exploratory orientation, concern for others, and emotion regulation) moderate the effect of cognitive components (knowledge, metacognitive capacities, and self-regulation) on wise behavior” (p. 342).

These three treatments require further study, but we will direct our readers to consider one more treatment that may not be widely recognized. It involves a contemporary rethinking of the ancient concept of practical wisdom (*phronesis* in the Greek).

One of the first important treatments of practical wisdom (*phronesis*) was by the psychologist Barry Schwartz and the political scientist Kenneth Sharpe ([89]) *Practical Wisdom*. They stress that to know “the right way to do the right thing,” as the subtitle of their book puts it, one must gain practical wisdom from the experience that one has in a “moral network” of parents, siblings, friends, neighbors, colleagues, and others with whom you interact. They stress that “you can’t make better teachers, doctors, and lawyers by simply telling them how to care for students, patients, and clients. They have to watch you doing it the right way, and you have to be correcting their mistakes, and tuning their networks, as they learn” (p. 102). Another highly regarded treatment of these matters is from [86]’s ([86]) *Practical Intelligence and the Virtues,* which is a welcome synoptic, critical, and thoughtful overview of the many issues involved in developing our understanding of practical intelligence (*phronesis*) and the virtues, and we suggest it is a worthwhile resource for anyone in this field of study.

But more recently there has been significant work by a group of scholars at the Jubilee Centre for Character and Virtues at the University of Birmingham (UK). We call this work to your attention because their work of rethinking and developing a concept of practical wisdom (*phronesis*) is so thorough and thoughtful, in particular by including the development and validation of an assessment instrument and by exploring the possibilities of developing *phronesis* as part of the ethics training for various professions, such as medicine, teaching, and policing (For more detail of their work, see *Phronesis: Retrieving Practical Wisdom in Psychology, Philosophy, and Education*, [62]). In their conception, *phronesis* is a complex, multi-sided psychological reality. It involves four distinct and interacting components, the first of which is the constitutive component. Succinctly put, “*[p]hronesis* involves the cognitive discriminatory ability to perceive the ethically salient or central aspects of a situation and to appreciate these as calling for specific kinds of responses” (p. 37). They allow that this component can be thought of as moral sensitivity or moral perception. Among other things, this component requires (a) the possession of concepts like gratitude, courage, and many other virtues, as well as (b) the imagination to envision a variety of possible actions. Kristjansson and Fowers acknowledge that the *phronimos*, the person possessing *phronesis*, will often, when presented with a moral dilemma, be able to see beyond the options presented to imagine new responses that may resolve the dilemma.

The second component is emotional regulation. In their view, the practically wise *phronimos* will feel emotions in reaction to a situation that motivates making a decision, but will reflect on whether the emotions are appropriate and proportionate. In their view, emotions are not so much suppressed by reason, but rather emotions are *infused* with reason.

The third component is one that there is controversy about, and they call it the blueprint component. Here is how they describe it:

The agent’s own ethical aims and aspirations, her understanding of what it takes to live and act well, and her need to live up to the standards that shape and are shaped by her understanding and experience of what matters. This amounts to what we call a blueprint of flourishing. ([62])

This need not be too obscure if we allow that there is likely to be a contrast in ethical decision-making between a police officer who knows that they want to be a good and effective officer versus one who simply enjoys exercising power over their fellow citizens.

Finally, the fourth component is the one that brings things together and thus is the integrative component. They ask you to imagine, for example, a situation in which being an honest friend might move you to tell your friend about the life-long unfaithfulness of their partner, but your friend is terminally ill, and compassion might urge you to say nothing. As they describe this component:

[A]n individual integrates different virtue relevant considerations, via a process of checks and balances, especially in circumstances where different ethically salient, or different kinds of virtues or values, appear to be in conflict and agents need to negotiate a dilemmatic space. (pp. 44–45)

The four components do interact with each other, but especially from a developmental standpoint they also appear to have a degree of independence which allows for some of the components to be more fully present than the others. This brief summary can hardly do justice to the richness and nuance of their account, and, in particular, their discussion in Chapter 6 of the research and validation efforts that have gone into constructing an assessment instrument is, to us, required reading for anyone who wishes to pursue the topic.

Before we leave this topic, there is one suggestion that seems to us to be worth making. It strikes us that, as comprehensive and thoughtfully developed as the above account is, there is an aspect of practical wisdom that should perhaps be highlighted more. In [89]’s ([89]) treatment of practical wisdom, they note early in their account:

Practical wisdom demands more than the skill to be perceptive about others. It also demands the capacity to perceive oneself—to assess what our own motives are, to admit our failure, to figure what has worked or not and why. (pp. 24–25)

Thus, the person who has practical wisdom both understands themselves well and knows how to “read” people accurately. These abilities are crucial, for, without them, decisions about what to do or to recommend will likely go awry. Do notice, however, that some people may have a keen ability to read others, but, if their “blueprint” includes something like “there’s a sucker born every minute for me to take advantage of,” their ability is not a component of practical wisdom in the sense being expounded here. As [62] ([62]) stress, a commitment to living a morally good life needs to be a salient part of the blueprint for these abilities to be part of practical wisdom.

### 5.3. For Wisdom-Building, Having a Socrates Figure as a Guide Is Nice, but Not Necessary

Paying attention to perceptions of oneself and others is reminiscent of [29]’s ([29]) use of the concepts of *intrapersonal intelligence* and *interpersonal intelligence.* Without getting embroiled in the controversies over his theory of multiple intelligences, it seems reasonable to think (a) that these abilities to perceive both oneself and others accurately do vary, with some people being very good at exercising one or both of these abilities while others do not do so well, and (b) that these abilities can be enhanced. The enhancement of these thinking abilities seems to be an exercise in what we earlier called learnable intelligence, fixing or improving the operation of the cognitive gadgets (the mindware) employed in these two domains.

Indeed, when we cited [100] ([100]) above, they allowed that this process would require a Socrates figure, but let us remember the story from John [33] ([33]) we recounted earlier about what it was that enabled the young man Mac, who was acting out in the juvenile corrections facility, to recognize flaws in his thinking. *It was peer-group counseling*. We suggest that there is an instructive structural parallel between that process and others that enable wisdom-building without the need for a Socrates figure. Instead, the processes of wisdom-building can take place among groupings of ordinary people, provided those ordinary people are willing and able to follow necessary guidelines that encourage honest and thoughtful communication.

The first example of such processes we bring to your attention is a specific instantiation of the Philosophy for Children (P4C) movement. This instantiation was in Scotland by Paul Cleghorn and Stephanie Baudet, who created the discussion materials *Thinking Through Philosophy*, *Vols. 1–4* ([14]). Teachers who used the materials were to follow a seven-stage lesson plan. The lessons began with:a focusing exercise to create a relaxed, meditative frame of minda brief link with the previous week’s discussionthen came the stimulus for discussion—usually reading a story from *Thinking Through Philosophy*next the students work in pairs discussing the story and reflecting on some open-ended questions suggested by the storythis was followed by dialogue in larger groups where the teacher has encouraged the students to form a *community of inquiry* by
(a)communicating their views in response to the questions at hand,(b)supporting their views with reasons,(c)listening respectfully to the views of others,(d)indicating whether they agree or disagree with those views,(e)providing alternative viewpoints, and(f)gradually developing a process of dialogue.the teacher brings closure by encouraging the students to reflect on the discussion and how their thinking might have progressed by providing a “thought for the week” that highlights an idea to serve as a basis for “homework” to be reflected on in order to relate the idea to situations outside the original stimulus ([26]).

The effects of this process were studied by Keith Topping and Steven Trickey ([103], [104]; [105]), who found improvement among the experimental group students over the control group students in their scores on the Cognitive Abilities Test (CAT in the UK, CogAT in the USA); even better, these results were still visible in a follow-up study performed two years later.

The results were so striking that an attempt to replicate their controlled experiment was undertaken in a medium-sized intermediate school in Texas. It turned out that the improvements in CogAT scores seen in Scotland were also observable among the seventh graders in Texas who were in the experimental group, and, what is striking, the improvements were still visible in a follow-up study performed three years later when the students were sophomores in high school ([25], [26]). Why does this matter? Because there is available an educational intervention which has serious evidence of its positive impact on students, and which is not expensive.

To expand on this last point, [25] ([25]) state that

Staff development time needed is minimal. One day of in-service sufficed.The materials are inexpensive. For example, at the beginning of 2013 a downloadable pdf copy of *Thinking Through Philosophy*, *Book 4*, can be purchased from Educational Printing Services in the UK for £17.50 (about $25.00 USD).The P4C lessons require only one hour per week of instructional time.The extent of time required for the program to have an effect is reasonable—25 weeks or so ([25]).

And here is a brief sample of some student comments about how the experience affected their approach to discussing issues with others:I learned how to show facts and find evidence and how to use it during disputes.It helped me understand that close mindedness is not a good thing.It helps you think about choices you make and view the world differently.I learned that to put your opinion out there you have to make sure you know what you are saying.I learned that you should give people a chance to talk because they might change your mind ([25]).

In recent provocative studies of the development of adolescents’ brains, researchers have noted that civic reasoning––“rigorously examining available evidence and multiple perspectives around meaningful personal and social issues”—has the capacity to transform neural networks. Thus, they argue that “[t]ransdisciplinary complementary evidence is accumulating that civic reasoning may be particularly developmentally powerful in part because of the effortful, emotionally invested, complex thinking it invokes—what we are calling *transcendent thinking*” ([50], emphasis in the original) (See also [49]).

A second example of the use of peer group discussion to reshape people’s mindware comes from an experiment in a college setting that adapted a pedagogical approach called Peer-Led Team Learning (PLTL), an approach first pioneered in an attempt to improve students’ performance in large university Introductory Chemistry classes ([38]). [45] ([45]) report on a research project that aimed to compare (a) the results of using a novel PLTL-inspired pedagogy with (b) the results of a standard method of instruction on the writing ability of students in one of the first college courses they take, namely Freshman Composition. The novel pedagogy consisted of two components: (1) reading texts like *Antigone* that present ethically charged situations, and (2) on a regular basis breaking the class up into a couple of smaller groups of about 10–12 students. The smaller groups would meet to discuss their reactions to prompts supplied by the instructor, prompts inspired by the texts but, importantly, *with the instructor not present* and, instead, with only a student leader whose job was just to take role and to monitor participation in the ensuing discussion to encourage everyone to participate.

When the results from three successive fall semesters were compared by blind scoring of final essays written by students in the PLTL experimental Freshman Composition sections and those written by students in the standard sections, the students in the experimental PLTL sections consistently and significantly outperformed their counterparts in the standard composition sections. Here are some student testimonials to convey the essence of what happened with many of the students:

(1) I will remember the lessons of Julius Caesar, Dr. Faustus, and Medea more than any other of the lessons. I will never let pride get the better of me.

(2) Although it did not start out that way, throughout this semester I have learned that I can develop a topic into a well-developed paper that I can be proud to display to others. I have also found out that people will always have different views and expectations throughout their lifetimes but with new insight and information, an individual can clearly see things in another persons’ point of view.

(3) It may sound funny, but I was pretty intimidated by a couple of students this semester. After PLTL I learned that you basically can’t judge a book by the cover.

(4) The stories we read, such as Julius Caesar, Medea, and Frederick Douglass, opened up my world, I learned so much from all of these stories.

This is a testimonial from an instructor:

My approach to the course has been one of highlighting individual moral development, so I encourage the students toward introspection and discussion with one another. Students responded well to hypothetical ethical dilemmas, and I asked them to analyze the rationale behind the moral decisions of literary characters. I was impressed by the level of camaraderie that the students showed by the end of the semester, and their willingness to hear multiple viewpoints.

In addition to practices like P4C and PLTL that take place in a formal educational setting, there are “how to” books that report ways in which practical wisdom can be enhanced by participation in other similarly structured processes. One example is *The Journey of Leadership: How CEOs Learn to Lead from the Inside Out* by [66] ([66]) of the global business consulting firm McKinsey. Another example is *How to Know a Person* by [9] ([9]). While the two books differ in many ways, there is one way they resemble each other and the P4C and PLTL practices, and that is in the way they describe the particular social structure of the process of seeking wisdom.

[66] ([66]) report on how CEOs participate in a special setting, which they call “the Bower forum,” named after a long-serving senior partner, a setting in which “McKinsey’s best practices in leadership are put to work, offering CEOs a proven approach to reinvent themselves” (p. ix). The forum brings together a small group of CEOs who face similar challenges in leading their organizations, otherwise extremely able people who were unable to “spark passion in their employees” and who “on a deep psychological level were not authentically connecting with themselves, and, equally important, with other members of their team” (p. ix).

The group operates under the Chatham House Rule, which specifies that “[w]hen a meeting, or part thereof, is held under the Chatham House rule, participants are free to use the information received, but neither the identity nor the affiliation of the speaker(s), nor any other participant, may be revealed.” The rule helps to “create a trusted environment to understand and resolve complex problems” (See https://www.chathamhouse.org/about-us/chatham-house-rule, accessed 18 June 2025). The Bower forum can function in much the same way as the P4C community of inquiry functions by enabling people to honestly exchange and evaluate ideas in a safe setting. They thereby not only improve their ability to respond to situations that raise challenging issues for them as executives, but also learn something about themselves and other people in the process. McKinsey’s term for the process is learning to lead “from the inside out.” The point is that, like P4C and PLTL, the Bower forum groups operate with relatively honest exchanges between peer CEOs. *These processes modify one’s ethical decision-making intelligence apparatus with the result that the people involved become smarter and wiser in responding to ethically charged situations*.

Similarly, in *How to Know a Person*, [9] ([9]) uses a number of vivid anecdotes, including one about a tsunami and its effects, to make his point, but in the last chapter, “What is Wisdom,” he admits that he has come to see that “wisdom is not mostly a trait possessed by an individual. Wisdom is a social skill practiced within a relationship or a system of relationships” (p. 263). Indeed, people can form a “community of truth”. That is:

A community of truth is created when people are genuinely interested in seeing and exploring together. They do not try to manipulate each other. They do not immediately judge, say “That’s stupid” or “That’s right.” Instead, they pause to consider what the meaning of the statement is to the person who just uttered it. (p. 264)

These communities can be as small as two people in conversation or might include groups in classrooms, or even on occasion the much larger entire enterprise of science with thousands of people spread out across the globe. Importantly, Brooks’ account reminds us that “communities of truth” do not require immediate, face-to-face interaction.

In summary, we think the important implication here about learnable intelligence and about wisdom-building processes, is that, contrary to what [100] ([100]) said, these processes do not need a hero-philosopher like Socrates to guide them. Incarnations of Socrates are few and far between, so it is good news that there are educational practices conducted with regular people that have been shown to have a degree of empirical support for the practices’ ability to advance participants’ knowledge and wisdom concerning ethically charged issues. We hope we have described the relevant research in sufficient detail so that the reader can decide whether to follow up by consulting the original sources.

## 6. Conclusions: So Where Do We Go from Here?

To conclude our consideration of the relation between intelligence and moral development, let us review our exploration of the research. As we do this, it needs to be understood that we do not claim to have settled definitively the matters we have addressed. But our central conclusion is that to understand deeply the relation between intelligence and moral development, as well as the effects of intelligence on moral development, requires more than the correlational studies that relate IQ scores to scores on an instrument purporting to measure moral development. Such studies are useful, no doubt, when conducted with the required methodological upgrades we have noted, but significant time and study must also be devoted to relevant topics, such as (1) how cognitive evolutionary psychology can help us understand the starting points of intellectual and moral development, (2) how we can grasp moral principles at an early age and construct mechanisms for moral decision making, (3) how has philosophy influenced the development of instruments used to assess moral development, and (4) how recent neuroscientific developments can deepen our understanding of these matters. So, what follows are suggestions for further study. 

### 6.1. Suggestions for Understanding Intelligence

One of the first inquiries is into the biological basis of intelligence, more specifically into intelligence in relation to brain development. For example, [16] ([16]) reported that “gray matter [in the brain] supports information processing capacity and white matter promotes efficient flow of information across the brain” (p. 497), both of which are important for intelligence. Research on brain development is more available because of the much-improved ways we can investigate the physiological aspects of intelligence ([97]).

Second, we recommend viewing intelligence in the light of [46]’ ([46]) cognitive evolutionary psychology which distinguishes a basic “starter set” of cognitive processes that then, very much based on social experiences, proceed to develop “cognitive gadgets” which enable us to cope with a variety of domains, including normative thinking. Then we noted some recent work by [90] ([90]) on how young, usually preschool, children’s imaginations are expanded in a variety of areas, including by the acquisition of prerequisites for learning moral principles. An example of one such prerequisite is the recognition that bad outcomes are not always caused by wrongdoing.

This guided us to the idea of “mindware,” the set of cognitive gadgets that we continue to develop over time in a number of areas, including the domain of morality. But as [95] ([95]) and others have noted, the pieces of mindware needed to reach correct conclusions in a variety of settings may be missing or defective. In the domain of morality, earlier in this paper, we illustrated this with the example of the antisocial behavior of the juvenile delinquent “Mac.” Mac’s inappropriate rage at being asked by a staff member to allow a search of his bag was seen, after peer group counseling, because of thinking errors such as “Assuming the Worst.” These considerations led us to suggest the notion, thanks to the work of [79] ([79]), of learnable intelligence.

### 6.2. Suggestions for Understanding Moral Development

We then considered how we might understand moral development. To begin, we studied the relationship between intelligence and moral development empirically, starting with one of the earliest such studies conducted by [113] ([113]) and then moving to more recent work such as that of [102] ([102]), [7] ([7]), and [87] ([87]). In this regard, we noted some problematic features, especially of earlier studies, and concluded with three specific suggestions about future directions this research might fruitfully follow:Using participants who are below the average range of intelligence (e.g., [7]) and noting that this brings up the issue that must be addressed of how best to measure intelligence,Including preschool children in one’s research (e.g., [7]), andConducting cross-cultural studies.

Then we made other suggestions for the research, one of which is to consider in depth the efforts of [30] ([30], [31]) to describe our common morality. Gert’s clarification of the nature of the goods and evils that the moral rules are about, his depiction of the relatively simple rules that forbid inflicting various evils, his calling attention to justified violations of the rules, and his recognition that morality also includes moral ideals that promote bringing about various good effects for people, all make us wonder what moral development would look like given such a framework. In this vein, we noted the work of [18] ([18]) with children 7 and 8 years of age who were unanimous in judging it to be wrong to deceive a teacher who commanded a student not to harm others, but a strong majority of the same children also judged it to be right to deceive a teacher who commanded one student to harm another. So, the young children both knew the relevant moral rule and understood that there could be a justifiable breaking of that moral rule, and this is what [30]’s ([30], [31]) depiction of morality would lead us to expect. For a similar stress on perceptions of harmfulness underlying moral thinking informed by decades of psychological and neuroscientific research, we recommended a careful look at Kurt Gray’s book *Outrage!* ([39]).

### 6.3. Suggestions About Nurturing Moral Intelligence

Another suggestion of work to consider is the recent developments in the study of practical wisdom (*phronesis*). Robert Sternberg and his colleagues (e.g., [100]) deserve great credit for pursuing the goal of making wisdom, which in their understanding is an inherently ethical matter, because it aims to attain the common good, and [89]’s ([89]) pioneering work on practical wisdom also should be noted, because they were among the first to stimulate interest in modernizing phronesis. However, to us the most intriguing work is that of the team at the Jubilee Centre for Character and Virtues. They have not only clarified the psychological structuring of practical wisdom *(phronesis)* as involving four key components––(1) moral sensitivity, (2) emotional regulation, (3) a “blueprint” of what it takes to live and act well, and (4) an ability to integrate different virtue relevant considerations––but they have also developed and validated an instrument, the Short Phronesis Measure (SPM), for assessing it ([62]; [63]). This research has elevated the concept of practical wisdom into something that can be the subject of empirical studies.

As striking as this work has been, we are concerned that there is an underemphasis on the intelligence abilities that we consider to be at the heart of practical wisdom: the need for accurate perception of others and oneself. As the case, cited earlier, of Mac and his peer group counseling suggests ([33]), mindware needed for good ethical decision making may be missing or flawed, and that led us to point to the existence of a number of social practices structurally similar to peer group counseling that can have a similar effect on participants by improving their relevant mindware. These practices include (1) the community of inquiry in a Philosophy for Children application, (2) a combination of Peer-Led Team Learning and works of literature which raise ethical issues for discussion in college freshman English classes, (3) the Bower forum gatherings of CEOs under Chatham House rules that enable open and honest discussion of issues of concern, and (4) [9]’ ([9]) depictions of how to come to know a person by participating in what he describes as “a community of truth.”

Lastly, we should note that it would be worthwhile pursuing a line of research regarding the influence of neurobiology on the relationship between intelligence and moral development. There has, of course, been research on the relationship between neuroscience (e.g., brain development) and intelligence, and there has been an increase in research that relates to the relationship between neuroscience and moral development (for a sample, see [49]).

It has been quite a journey, but we would like to believe that what we have presented here can provide useful avenues for our fellow explorers who are trying to come to a better understanding of the relationship between intelligence and moral development.

## Data Availability

No new data were created or analyzed in this study. Data sharing is not applicable to this article.

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
