# Peer review of "Intelligence and Moral Development: A Critical Historical Review and Future Directions"

_jintelligence, 2025, doi:10.3390/jintelligence13070072_

Round 1
Reviewer 1 Report
Comments and Suggestions for Authors
This manuscript addresses an underexplored and important intersection: intelligence and moral development. The authors employ rich references across disciplines (psychology, philosophy, neuroscience, education). The manuscript is generally well-written with appropriate academic tone and citation practices. The review of previous empirical studies is generally thorough.
Recommendations:
- The abstract is quite dense and tries to cover too many themes; a clearer structure would improve readability.
- The transitions between sections are sometimes abrupt. For instance, the leap from cognitive neuroscience to moral development could benefit from clearer marking.
- The discussion of older studies (e.g., Wiggam, 1941) is overlong and could be condensed in favor of more recent work.
- Authors should ensure any self-citation is relevant and proportionate.
Author Response
Reviewer #1 Comments and Replies.
This manuscript addresses an underexplored and important intersection: intelligence and moral development. The authors employ rich references across disciplines (psychology, philosophy, neuroscience, education). The manuscript is generally well-written with appropriate academic tone and citation practices. The review of previous empirical studies is generally thorough.
Recommendations:
- The abstract is quite dense and tries to cover too many themes; a clearer structure would improve readability.
--We redid the Abstract to clarify things. We also added an Introduction that specifies a three-part main structure followed by a Conclusion in which specific suggestion for further exploration are made. Here is the new Introduction:
This paper is a critical, historical review of the literature on the relationship between intelligence and moral development. It has three main parts: “Part I Intelligence from a “Starter Kit” to Mindware”, “Part II Intelligence and Moral Development: Multiple Perspectives, Classic Research, and a Different Take on Morality”, and “Part III Intelligence and Morality in Action: Wisdom and Wisdom-building.” The final section of the paper “Conclusions: So Where Do We Go from Here?” contains a number of suggestions of what seem to us to be fruitful avenues of exploration for anyone interested in the subject.
- The transitions between sections are sometimes abrupt. For instance, the leap from cognitive neuroscience to moral development could benefit from clearer marking.
--We added some subheadings at this juncture “A Psychological Perspective” “A Social Perspective” and “A Neuroscience Perspective” to make the organization clearer.
- The discussion of older studies (e.g., Wiggam, 1941) is overlong and could be condensed in favor of more recent work.
--We trimmed this section.
- Authors should ensure any self-citation is relevant and proportionate.
--We have made an effort to do so. The sources that are self-citations are few in number, four in fact, they are all relevant, and we think needed for our narrative.
Reviewer 2 Report
Comments and Suggestions for Authors
The author has included a wealth of information on moral development and has come at the topic from an array of perspectives. The problem I have is what seems to be an overload of information presented sequentially, sometimes with an inordinate amount of detail, but a lack of organization and integration that leaves the reader (at least this reader) a bit overwhelmed. And whereas the author has stated up front that they have not attempted to include all relevant works, it seems to me that they have failed to mention some that might be considered essential (I have indicated a few references below that I consider to be in that category by Haidt, Shweder, Iyer, Jayawickreme & Fleeson, Tanner & Christen). The author can’t be faulted for failing to include Sternberg’s most recent Review of General Psychology paper detailing his theory of moral development because it just appeared online. But now that Sternberg’s theory is available, it is incumbent on the author of the present paper to feature it, and address its key points, in their revision.
Haidt, J. (2013). The righteous mind: Why good people are divided by politics and religion. Knopf.
Iyer, R., Koleva, S., Graham, J., Ditto, P., & Haidt, J. (2012). Understanding libertarian morality: The psychological dis- poitions of self-identified libertarians. PLoS One, 7(8), Article e42366. https://doi.org/10.1371/journal.pone.0042366
Jayawickreme, E., & Fleeson, W. (2017). Does whole trait theory work for the virtues? In W. Sinnott-Armstrong & C. B. Miller (Eds.), Moral psychology: Virtue and character (pp. 75–103). Boston Review. https://doi.org/10.2307/j.ctt1n2tvzm.9
Shweder, R. A., Much, N. C., Mahapatra, M., & Park, L. (1997). The ‘big three’ of morality (autonomy, community, divinity) and the ‘big three’ explanations of suffering. In A. M. Brandt & P. Rozin (Eds.), Morality and health (pp. 119–169). Taylor & Frances/Routledge.
Tanner, C., & Christen, M. (2014). Moral Intelligence—a frame- work for understanding moral competences. In M. Christen, C. van Schaik, J. Fischer, M. Huppenbauer, & C. Tanner (Eds.), Empirically informed ethics: Morality between facts and norms (pp. 119–136). Springer International Publishing/Springer Nature. https://doi.org/10.1007/978-3-319-01369-5_7
Robert J Sternberg A Trilogy Theory of Moral Intelligence. Review of General Psychology 2025, Vol. 0(0) 1–19
Article reuse guidelines: sagepub.com/journals-permissions DOI: 10.1177/10892680251331852 journals.sagepub.com/home/rgp
Author Response
Reviewer #2 Comments and Replies
The author has included a wealth of information on moral development and has come at the topic from an array of perspectives. The problem I have is what seems to be an overload of information presented sequentially, sometimes with an inordinate amount of detail,
--We reduced the level of detail, but since we are trying to give readers some guidance about what we hope are fruitful directions for further research, we felt it necessary to include sufficient detail so that readers could judge for themselves whether a given direction was likely to be fruitful. For example, we think Heyes’ notion of a starter set of cognitive processes is an intriguing idea that needs further exploration, such as pondering exchanges between Heyes and others about whether violations of “core knowledge” play an essential role in early learning.
but a lack of organization and integration that leaves the reader (at least this reader) a bit overwhelmed.. --We made the organization of the article clearer. We deleted some less than essential discussions, and we made the structure clearer by noting that the paper has three distinct parts and a conclusion that offers a number of suggestions.
And whereas the author has stated up front that they have not attempted to include all relevant works, it seems to me that they have failed to mention some that might be considered essential (I have indicated a few references below that I consider to be in that category by Haidt, Shweder, Iyer, Jayawickreme & Fleeson, Tanner & Christen). The author can’t be faulted for failing to include Sternberg’s most recent Review of General Psychology paper detailing his theory of moral development because it just appeared online. But now that Sternberg’s theory is available, it is incumbent on the author of the present paper to feature it, and address its key points, in their revision.
--Thanks to Reviewer #2 for these suggestions. We are consulting them all, and we have incorporated several of them into the article. The recent Sternberg piece definitely deserves notice, for example.
Haidt, J. (2013). The righteous mind: Why good people are divided by politics and religion. Knopf
--We have incorporated a brief discussion of Haidt’s views and those of a critic, Kurt Gray
Iyer, R., Koleva, S., Graham, J., Ditto, P., & Haidt, J. (2012). Understanding libertarian morality: The psychological dis-positions of self-identified libertarians. PLoS One, 7(8), Article e42366. https://doi.org/10.1371/journal.pone.0042366
--This appears not to be directly relevant to our project since it is about the psychology of a particular political orientation. We would prefer not to introduce explicitly political considerations into our work.
Jayawickreme, E., & Fleeson, W. (2017). Does whole trait theory work for the virtues? In W. Sinnott-Armstrong & C. B. Miller (Eds.), Moral psychology: Virtue and character (pp. 75–103). Boston Review. https://doi.org/10.2307/j.ctt1n2tvzm.9
--While an interesting piece, its focus on conceptualizing the virtues seems tangential to the main thrust of our paper.
Shweder, R. A., Much, N. C., Mahapatra, M., & Park, L. (1997). The ‘big three’ of morality (autonomy, community, divinity) and the ‘big three’ explanations of suffering. In A. M. Brandt & P. Rozin (Eds.), Morality and health (pp. 119–169). Taylor & Frances/Routledge.
--We incorporated a brief reference to Shweder et al.’s work.
Tanner, C., & Christen, M. (2014). Moral Intelligence—a frame- work for understanding moral competences. In M. Christen, C. van Schaik, J. Fischer, M. Huppenbauer, & C. Tanner (Eds.), Empirically informed ethics: Morality between facts and norms (pp. 119–136). Springer International Publishing/Springer Nature. https://doi.org/10.1007/978-3-319-01369-5_7
--We have incorporated a discussion of this work into the paper.
Robert J Sternberg A Trilogy Theory of Moral Intelligence. Review of General Psychology 2025, Vol. 0(0) 1–19
Article reuse guidelines: sagepub.com/journals-permissions DOI: 10.1177/10892680251331852 journals.sagepub.com/home/rgp
--We have incorporated a discussion of this into the paper.

Reviewer 3 Report
Comments and Suggestions for Authors
Review of Intelligence and Moral Development Paper
- Many problems in life don’t have a “right answer.” They may not even have a good answer.
- The problem with the Stanovich approach of intelligence is what IQ tests measure is that it is atheoretical and operationist. It tells us nothing about what intelligence is or how it operates. In the early 20th century, maybe we could not do better, but it is a century later.
- Isn’t all mindware flawed? If you look at people’s real thinking in the everyday world, they seem to operate at a much lower level than one might hope for, given performance on standardized tests.
- Sometimes, flawed mindware may be good if it leads us to creative thinking rather than just processing what we have as everyone else does.
- The Haier-Jung neuropsychological model P-FIT seems not to be cited. The authors should check for how up-up-do-date their citations are. They may have missed some recent work. For example, relevant uncited papers would be:
Grossmann, I. Weststrate, N. M., Ardelt, M., Brienza, J. P., Dong, M., Ferrari, M., Fournier, M. A., Hu, C. S., Nusbaum, H. C., & Vervaeke, J. (2020). The science of wisdom in a polarized world: Knowns and unknowns. Psychological Inquiry, 31(2), 1–31. doi: 10.1080/1047840X.2020.1750917
Glück, J., & Weststrate, N. (2022). The wisdom researchers and the elephant: An integrative theory of wise behavior. Social and Personality Psychology Review, 26(4), 342-374. https://doi.org/10.1177/10888683221094650
Sternberg, R. J. (2025). A trilogy theory of moral intelligence. Review of General Psychology. https://doi.org/10.1177/10892680251331852
Sternberg, R. (2024). What is wisdom? Sketch of a TOP (tree of philosophy) theory. Review of General Psychology, 28(1), 47-66. https://doi.org/10.1177/10892680231215433.
6. Where did Gert’s list come from? Did he make it up? Doctors and dentists sometimes cause pain, for good causes. Doctors may disable when they have to amputate a limb. Some laws are immoral and should not be obeyed, especially in malign autocracies. The list seems somewhat arbitrary.
- The paper seems to wander somewhat. Does it have a clear structure?
- Is there any thesis that the paper is trying to prove, or set of theses? The paper might be improved by having a clear thesis and structure.
- Are intelligence abilities really at the heart of practical wisdom? They don’t seem to work for all our Ivy-educated Congress people in the US. A lot of academically brilliant people are duds in practical matters.
- What exactly are the conclusions we are to draw from all this?
- The paper does read like a journey but it needs to give us more of a clear path that it follows, maybe with an advance organizer.
- The paper would profit from saying, in the end, what can be concluded about moral development and intelligence. I’m not sure I see what it is.
Author Response
Reviewer #3 Comments and Replies
Review of Intelligence and Moral Development Paper
- Many problems in life don’t have a “right answer.” They may not even have a good answer.
--Our article makes no claim that all problems have a demonstrably correct answer. Indeed, it is one of the strengths of Bernard Gert’s depiction of morality that it allows for situations where well-intentioned, thoughtful people can rationally disagree.
- The problem with the Stanovich approach of intelligence is what IQ tests measure is that it is atheoretical and operationist. It tells us nothing about what intelligence is or how it operates. In the early 20th century, maybe we could not do better, but it is a century later.
--Please note that our focus, like the title of Stanovich’s book, is on What Intelligence Tests Miss. Stanovich adopts the term “rationality” for this, but we prefer to look at it as a matter of mindware, that is, learnable intelligence.
- Isn’t all mindware flawed? If you look at people’s real thinking in the everyday world, they seem to operate at a much lower level than one might hope for, given performance on standardized tests.
--The term “mindware” is an analogy, of course, with computer programs. Thus, a distinct piece of mindware might be thought of as a particular subroutine or component of the larger package. See the example we give of learning a piece of mindware by becoming aware of the fallacy of “base rate neglect” in situations involving estimation of the probative impact of a positive medical test result. Surely some pieces of mindware function better than others, and that is why the story of Mac and his recurrent moral thinking errors being identified and challenged by members of his peer group counselors is relevant.
- Sometimes, flawed mindware may be good if it leads us to creative thinking rather than just processing what we have as everyone else does.
--It is hard to respond to this without an example. It is obvious that there can be “happy accidents.” What we are concerned with is reducing systematic gaps and proneness to errors that are harmful.
- The Haier-Jung neuropsychological model P-FIT seems not to be cited. The authors should check for how up-up-do-date their citations are. They may have missed some recent work.
--We have incorporated a discussion of Jung and Heier’s 2007 target article, The Parieto-Frontal Integration Theory (P-FIT) of intelligence: converging neuroimaging evidence from Behavioral and Brain Sciences, 30, 135-187.
- For example, relevant uncited papers would be:
--Thanks for Reviewer #3 for these helpful references.
Grossmann, I. Weststrate, N. M., Ardelt, M., Brienza, J. P., Dong, M., Ferrari, M., Fournier, M. A., Hu, C. S., Nusbaum, H. C., & Vervaeke, J. (2020). The science of wisdom in a polarized world: Knowns and unknowns. Psychological Inquiry, 31(2), 1–31. doi: 10.1080/1047840X.2020.1750917
--We have incorporated a reference to this into the paper.
Glück, J., & Weststrate, N. (2022). The wisdom researchers and the elephant: An integrative theory of wise behavior. Social and Personality Psychology Review, 26(4), 342-374. https://doi.org/10.1177/10888683221094650
--We have incorporated a reference to this into the paper.
Sternberg, R. J. (2025). A trilogy theory of moral intelligence. Review of General Psychology. https://doi.org/10.1177/10892680251331852
--We have incorporated a reference to this in the paper.
Sternberg, R. (2024). What is wisdom? Sketch of a TOP (tree of philosophy) theory. Review of General Psychology, 28(1), 47-66. https://doi.org/10.1177/10892680231215433
--We have incorporated this in the paper.
- Where did Gert’s list come from? Did he make it up?
--In Gert’s depiction the point of the moral rules is to lessen the unjustified infliction of harms. See his discussion of goods and evils that does not try to specify an ultimate good, but instead focuses on things that aid or hinder us in accomplishing our goals. If, for example, being killed ordinarily defeats your ability to accomplish your goals, then being killed is being harmed. And so on for the many other harms. Hence one of the moral rules is “Don’t kill.”
Doctors and dentists sometimes cause pain, for good causes. Doctors may disable when they have to amputate a limb.
--It is one of the strengths of Gert’s description of the moral rules that it explicitly incorporates a specific process for justifying such exceptions to the rules.
Some laws are immoral and should not be obeyed, especially in malign autocracies. The list seems somewhat arbitrary.
--Indeed, we cite a source to that same effect on p. 14:
Rules place restrictions on you and add to your obligations that you should obey, but as Wagner et al. point out, “sometimes you will be asked to help challenge these duties and obligations”
- The paper seems to wander somewhat. Does it have a clear structure?
--This new Introduction should give the reader a better idea of the basic structure:
Introduction
This paper is a critical, historical review of the literature on the relationship between intelligence and moral development. It has three main parts: “Part I Intelligence from a “Starter Kit” to Mindware”, “Part II Intelligence and Moral Development: Multiple Perspectives, Classic Research, and a Different Take on Morality”, and “Part III Intelligence and Morality in Action: Wisdom and Wisdom-building.” The final section of the paper “Conclusions: So Where Do We Go from Here?” contains a number of suggestions of what seem to us to be fruitful avenues of exploration for anyone interested in the subject.
- Is there any thesis that the paper is trying to prove, or set of theses? The paper might be improved by having a clear thesis and structure.
--The paper has no singular thesis, but we think of it as a map of the territory wherein we point out directions that we think are worthy of further exploration and directions that appear to be dead ends.
- Are intelligence abilities really at the heart of practical wisdom? They don’t seem to work for all our Ivy-educated Congress people in the US. A lot of academically brilliant people are duds in practical matters.
--We agree of course. Academically brilliant people have a share of sophia, the kind of theoretical understanding of matters whether it be economics, literary criticism, world history, etc. that is the stock in trade of academics. That is why our focus is on practical wisdom, aka phronesis. In our paper two intelligence abilities that are integral elements of practical wisdom are interpersonal and intrapersonal intelligence.
Here is what our paper says on p. 45:
In Schwartz and Sharpe’s (2010) treatment of practical wisdom they note early in their account
Practical wisdom demands more than the skill to be perceptive about others. It also demands the capacity to perceive oneself—to assess what our own motives are, to admit our failure, to figure what has worked or not and why. (pp. 24-25)
Thus, the person who has practical wisdom both understands themselves well and knows how to “read” people accurately. These abilities are crucial for, without them, decisions about what to do or to recommend will likely go awry.
- The paper does read like a journey but it needs to give us more of a clear path that it follows, maybe with an advance organizer.
--See the new Introduction that should help the traveler setting out on the journey.
- The paper would profit from saying, in the end, what can be concluded about moral development and intelligence. I’m not sure I see what it is.
--Since there is no singular thesis, there are instead a number of points that we feel are significant and should be taken into account by anyone interested in the topic. For example, that intelligence can be best understood as a combination of a “starter kit” of cognitive processes giving rise in the normal course of development to many, many “cognitive gadgets” some of which deal with moral matters, as in mindware for moral intelligence. Or that past research in this area, while interesting, is rife with specific methodological issues that need to be dealt with. Etc.

Round 2
Reviewer 1 Report
Comments and Suggestions for Authors
The revised abstract is clearer and more focused.
The insertion of subheadings clarifies the shift between frameworks.
Thank you for condensing this section.
The authors have made meaningful revisions that improve the manuscript's clarity and academic rigor.
Author Response
Reviewer #1 Round 2 comments and replies
Comments and Suggestions for Authors
- The revised abstract is clearer and more focused.
--We are glad to hear that the revision does a better job.
- The insertion of subheadings clarifies the shift between frameworks.
--We hoped that it would have that effect, so thank you.
- Thank you for condensing this section.
--Thanks for noticing the effort.
- The authors have made meaningful revisions that improve the manuscript's clarity and academic rigor.
--This is music to our ears!
Reviewer 2 Report
Comments and Suggestions for Authors
The authors provide an amazing wealth of information on moral development and intelligence, from a remarkable diversity of perspectives. In this superb revision they have provided an excellent framework to organize the article and have effectively and articulately integrated the important references I suggested. They also eliminated some of the detail that distracted rather than enhanced. It is still a tough read because it is so power packed with information, but the authors are clearly authoritative on the topic and it will provide a great resource for anyone who conducts research on moral development and intelligence. And their future directions section offers valuable paths for researchers to take.
Author Response
Reviewer #2 Round 2 comments and replies
The authors provide an amazing wealth of information on moral development and intelligence, from a remarkable diversity of perspectives. In this superb revision they have provided an excellent framework to organize the article and have effectively and articulately integrated the important references I suggested.
--Thank you for your Round 1 comments and suggestions, and we are glad that our revisions in response to those comments did the job.
They also eliminated some of the detail that distracted rather than enhanced. It is still a tough read because it is so power packed with information, but the authors are clearly authoritative on the topic and it will provide a great resource for anyone who conducts research on moral development and intelligence.
And their future directions section offers valuable paths for researchers to take.
--It was our hope to accomplish exactly that, so thank you for your kind words.
Reviewer 3 Report
Comments and Suggestions for Authors
The topic is great. The author cites a lot of work. This should become a publishable paper. It is not there.
The article is better. But it still needs work.
1. Sometimes the paper is very superficial. For example, it now cites the Sternberg (2025) paper but shows no sign that the author has read it. And that paper is not a theory of wisdom but rather of moral intelligence, so it is mischaracterized. Ditto for the paper on TOP model of wisdom. There is no sign that the author read it. The depictions of Grossmann et al. and Glück and Weststrate are also very superficial, and not well tied in with the topic of the paper as a whole, morality and intelligence. So then I worry about the rest of the paper. Has the author read the work or just cited it?
2. The organization of the paper is not very tight. It wanders from topic to topic but does not really develop a clear thesis. Rather, it just talks about topics with showing how they illustrate some major thesis.
3. Having a thesis or underlying theory or theoretical framework would make this a better paper. What is the author trying to show?
4. What can be concluded, at the end, about what the author was trying to show? What is the take-home message? What have we learned?
Author Response
Reviewer #3 Round 2 comments
The topic is great. The author cites a lot of work. This should become a publishable paper. It is not there. The article is better. But it still needs work.
--We appreciate the encouragement, and we hope to have moved things in the right direction.
- Sometimes the paper is very superficial. For example, it now cites the Sternberg (2025) paper but shows no sign that the author has read it. And that paper is not a theory of wisdom but rather of moral intelligence, so it is mischaracterized.
--In response we have added a more developed account of Sternberg (2025), including noting the fact that Sternberg cited Aristotle’s Nicomachean Ethics. We’d like to think that “moral intelligence”, as Sternberg defines it, with it having three components, is very close to what Aristotle would call “practical wisdom” or phronesis. Thus “moral intelligence” is defined on the first page as “the knowledge, abilities and attitudes needed to apply universal principles of right and wrong to personal, collective, and societal problems.” There are significant challenges to Sternberg’s development of the trilogy theory, one of which is the lack of a compelling justification for selecting the particular universal principles determined to be part of the trilogy. But see the more extensive treatment that is now in the manuscript.
- Ditto for the paper on TOP model of wisdom. There is no sign that the author read it.
--In response we have added a more developed account of Sternberg (2024). The taxonomic endeavor Sternberg undertakes, while interesting, we believe suffers from some difficulties, two of which we note in our response, and its effort seems tangential to what we are doing. With regard to Sternberg and Glück (2022), on the other hand, we give an extensive account, in part because it leads nicely to a consideration of recent work on phronesis.
- The depictions of Grossmann et al. and Glück and Weststrate are also very superficial, and not well tied in with the topic of the paper as a whole, morality and intelligence.
--What we tried to do, in what is already a very long paper, is to present just enough detail to possibly pique the curiosity of an interested reader. The sheer breadth of the topic prevents going into every citation in the same depth.
- The organization of the paper is not very tight. It wanders from topic to topic but does not really develop a clear thesis. Rather, it just talks about topics with showing how they illustrate some major thesis.
- Having a thesis or underlying theory or theoretical framework would make this a better paper. What is the author trying to show?
--In response to Comments 4 and 5, we did not begin our exploration with a “major thesis” in mind. The point was to explore the relevant terrain to see what emerged as promising ways forward. We operated under the guidance of an assumption that that was a function of a Review paper which is what this is.
- What can be concluded, at the end, about what the author was trying to show? What is the take-home message? What have we learned?
--As a response to this and the previous two comments, we revised part of the Abstract and the Conclusion in an effort to make clearer the future avenues of inquiry that we uncovered in our exploratory journey.
Round 3
Reviewer 3 Report
Comments and Suggestions for Authors
This version is better than the previous one. What I find troubling, however, is that I can't figure out what the point of the paper is. The authors say in their reply to the reviewers that the paper has no point. This is an unusual statement. Generally, I think it is reasonable for professional papers to have a point and an organizing principle, not just be a series of observations. This paper, as it stands, is a series of observations. I think it would be a much better paper if it had a point to make. Perhaps wherever the authors live, papers do not need to have a point. But I think in most of science, papers are written to make a point and then prove a point, not just have a series of observations. As it stands, I think the authors shoot themselves in the foot when they say there is no point to the paper. They can and should do better. Hopefully, they will. The paper needs an organizing and take-home message.
Author Response
Reviewer #3 Round 3 Comments and Replies
Date of this review
04 Jun 2025 15:10:00
- This version is better than the previous one.
--We appreciate the recognition of steps taken to improve the article in response to Reviewer #3’s previous comments.
- What I find troubling, however, is that I can't figure out what the point of the paper is. The authors say in their reply to the reviewers that the paper has no point. This is an unusual statement.
--Perhaps there is some ambiguity here? We never said that “the paper has no point.” What we indicated is that it has no single thesis, no single statement for which evidence pro and con is presented. But, of course, the paper has a point in the sense of a purpose or an aim, namely to be, as we say in the Introduction:
“This paper is a critical, historical review of the literature on the relationship between intelligence and moral development.”
- Generally, I think it is reasonable for professional papers to have a point and an organizing principle, not just be a series of observations. This paper, as it stands, is a series of observations. I think it would be a much better paper if it had a point to make. Perhaps wherever the authors live, papers do not need to have a point, not just have a series of observations. As it stands, I think the authors shoot themselves in the foot when they say there is no point to the
paper.
--Perhaps we should say in response to this comment that we took seriously the description on p. 2 of the “Instructions for Authors” about papers that present reviews:
“Reviews offer a comprehensive analysis of the existing literature within a field of study, identifying current gaps or problems. They should be critical and constructive and provide recommendations for future research. No new, unpublished data should be presented.”
- They can and should do better. Hopefully, they will. The paper needs an organizing and take-home message.
--We think that in good faith we fulfilled the duties of authors of Reviews, as differentiated from Articles describing “scientifically sound experiments” (Instructions to Authors, p. 2) Notwithstanding that, however, we have created a revision of the Abstract that is descriptive of the multiple conclusions we arrived at in the course of doing the review.
: